# SkillNet: Hierarchical Skill Modeling for Compositional Generalization in Vision-Language Action Models

Senwei Xie [1 2]  Yuntian Zhang [1 2]  Zhenzhou Tan [1 2]  Ruiping Wang [1 2]
Pengwei Wang [3]  Shanghang Zhang [4]  Xilin Chen [1 2]

## Abstract

Transfer across diverse task compositions and unseen behaviors remains a significant challenge for vision-language action (VLA) models. Skills are repeatable and atomic components for various tasks, and similarities shared with different skills provide evidence for transferability across behaviors. However, existing skill-centric methods have two problems. First, skills are often loosely organized, lacking a hierarchy that can capture similarities and differences across skills. Second, they lack a mechanism which has the capacity to express transferable skill attributes in a structured parametric space. To this end, we propose SkillNet, which models skill attributes in a hierarchical way and regulates compositional model structure with transferable skill attributes. SkillNet exploits motion code and VerbNet Framework to explicitly model similarities of skills on mechanical properties and semantic roles, and organizes skills in a hierarchical way. Based on this hierarchy, SkillNet leverages the scalability of the Mixture-of-Experts (MoE) mechanism and develops skill embeddings as soft constraints to enable compositional generalization via similar expert activations on similar skills. On zero-shot and few-shot transfer experiments in simulators and real-world environments, SkillNet achieves an improvement of performance by 16.0% and 23.9%. Meanwhile, SkillNet achieves the state-of-the-art performance on in-domain settings.

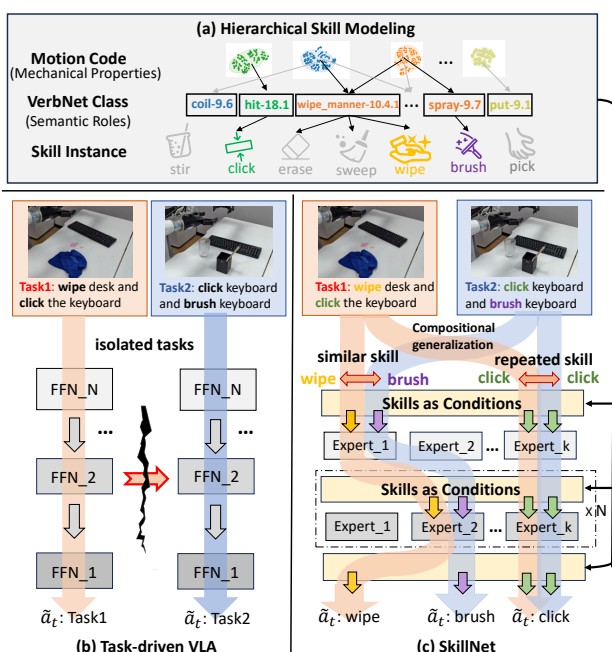

*Figure 1.* An overview of SkillNet. (a) SkillNet organizes skill entities with hierarchical skill modeling (HSM), considering similarities on mechanical and semantic attributes. (b) Task-driven VLAs learn task by task, overlooking transferable components in different tasks. (c) SkillNet exploits repeated skill components and similar skill attributes to transfer on diverse task compositions. Hierarchical skill embeddings serve as bottlenecks on MoE routing, formulating transferable parametric compositions.

[1]Key Laboratory of AI Safety of CAS, Institute of Computing Technology, Chinese Academy of Sciences [2]University of Chinese Academy of Sciences [3]Beijing Academy of Artificial Intelligence [4]State Key Laboratory of Multimedia Information Processing, School of Computer Science, Peking University. †Project lead: Ruiping Wang. Correspondence to: Ruiping Wang <wangruiping@ict.ac.cn>, Shanghang Zhang <shanghang@pku.edu.cn>.

*Proceedings of the 43rd International Conference on Machine Learning*, Seoul, South Korea. PMLR 306, 2026. Copyright 2026 by the author(s).

## 1. Introduction

Compositional generalization, which means generalization capabilities on different task compositions and unseen behaviors, is a long term goal for robotic manipulation. Vision-language action models (Kim et al., 2024; O'Neill et al., 2024; Black et al., 2024) attempt to reproduce scaling laws in natural language processing with larger scale trajectory training data. Despite better in-domain performance and generalization to novel objects, they still struggle on adaptation to unseen behaviors and task compositions. Recently, skill-centric methods (Liang et al., 2023; Huang et al., 2022; Wan et al., 2024; Liang et al., 2024), which treat tasks as compositions of atomic skills, provide a potential support

for compositional generalization. First, repeatable skills are atomic components of long-horizon tasks. Treating different tasks as compositions of atomic skills encourages policies to capture behavior patterns shared across tasks, instead of learning each task in isolation. Second, different skills share similar attributes on mechanics and semantic roles. When adapting to behaviors unseen during training, these shared structures on skill relationships provide a basis for transferring knowledge from previously established skills.

However, there are two problems to be addressed before effectively leveraging transferable properties of skills in learnable robotic policies. On the one hand, similarities and differences among skills should be explicitly characterized. These similarities range from mechanics properties (e.g., contact, deformable, constraints), semantic roles (e.g., target, context, instrument) to specific skill implementations, requiring a well-organized hierarchy to represent and summarize attributes at different abstraction levels. On the other hand, the policy model should have the capacity to leverage similarities characterized by the skill hierarchy, which can be implemented with a compositional parametric space conditioned by intermediate skill representations.

To this end, we propose SkillNet, a skill-centric framework with hierarchical skill modeling (HSM) and skill-contextualized Mixture-of-Experts mechanism for robotic policies. As shown in Figure 1 (a), HSM provides a structured organization of multi-grained skill attributes. First, HSM utilizes clusters of motion codes (Paulius et al., 2020) on heterogeneous demonstrations to summarize motion patterns shared across various skills. Clusters of these digital codes provide measurable abstractions of unstructured motions, encoding mechanic attributes on contact, trajectory type, and constraints. Complementing this, HSM further leverages VerbNet Framework (Schuler, 2005) to summarize semantic categories. While motion patterns capture similarities in mechanics, semantic categories are more specific to common attributes on targets, action semantics, and contextual roles. The resulting HSM forms a directed acyclic graph (DAG), progressing from motion categories, semantic categories, to fine-grained skill instances.

To better exploit the structured knowledge encoded in hierarchical skills, a robotic policy should have the capacity to express multi-grained skill relationships. As shown in Figure 1 (b), task-driven VLAs learn task by task in the same parametric space, overlooking shared or transferable components across tasks. We observe that the Mixture-of-Experts (MoE) architecture offers a promising alternative due to its inherent modularity and conditional activation capabilities, and then we propose Skill-Contextualized Mixture-of-Experts (SCMoE) to explicitly model skill similarities at the network level. As shown in Figure 1 (c), hierarchical skills are explicitly used to condition expert routing,

while stacking selected experts across MoE layers induces diverse activation pathways to express different skills. This design formulates a compositional parametric space, which enables structural sharing across similar skills. Specifically, when encountering unseen behaviors, SCMoE leverages the shared expert combinations of similar skills to facilitate rapid transfer through established behavioral patterns. For generalization to unseen task compositions, SCMoE models temporal transitions between skills, decomposing task sequences into previously learned skills to support compositional generalization on the task level.

To comprehensively evaluate the effectiveness of SkillNet, we designed three groups of different experiments, focusing on **few-shot transfer capability** to unseen behaviors, **zero-shot generalization** to unseen task compositions, and **in-domain** performance. All experiments are implemented on both real-world and simulation benchmarks. SkillNet demonstrates an improvement of 23.9% against baselines under few-shot transfer from basic skills to previously unseen behaviors. Under zero-shot condition on our proposed benchmark **LIBERO-Skill**, SkillNet significantly outperforms baseline methods by 16.0%. Finally, SkillNet also achieves consistent improvements on in-domain tasks, indicating that the proposed design remains beneficial under standard training settings. The routing analysis, including representational similarity analysis (RSA) and visualizations of router transitions, provides direct evidence of how skill hierarchies influence expert compositions and offers insights into the model's internal interpretability.

## 2. Related Works

### 2.1. Vision-Language Action Models

Built on large-scale robotic trajectory data (Khazatsky et al., 2024; O'Neill et al., 2024) and large multi-modal models, vision-language action models achieve outstanding performance across various in-domain robotic tasks (Brohan et al., 2023; Kim et al., 2024; 2025). Existing VLA models primarily follow two modes of action output: Auto-Regressive (AR) which output action tokens directly, and Flow-Matching methods which utilize diffusion heads to parallel decode action horizons. AR methods (Kim et al., 2024) are more efficient on training data scaling, and Flow-Matching methods (Black et al., 2025; 2024) benefit from parallel decoding to get higher inference frequency. To further scale up while preserve real-world control frequency of VLA systems, recent advances use a group of Mixture-of-Experts (MoE) adapters to extend existing VLA backbones (Zhou et al., 2025; Shen et al., 2025; Yang et al., 2025; Zhang et al., 2026). Despite better in-domain performance, generalization and transfer ability of VLAs to unseen behaviors or task compositions are still challenging, which calls for a systematic view on data, architecture, and evaluation.

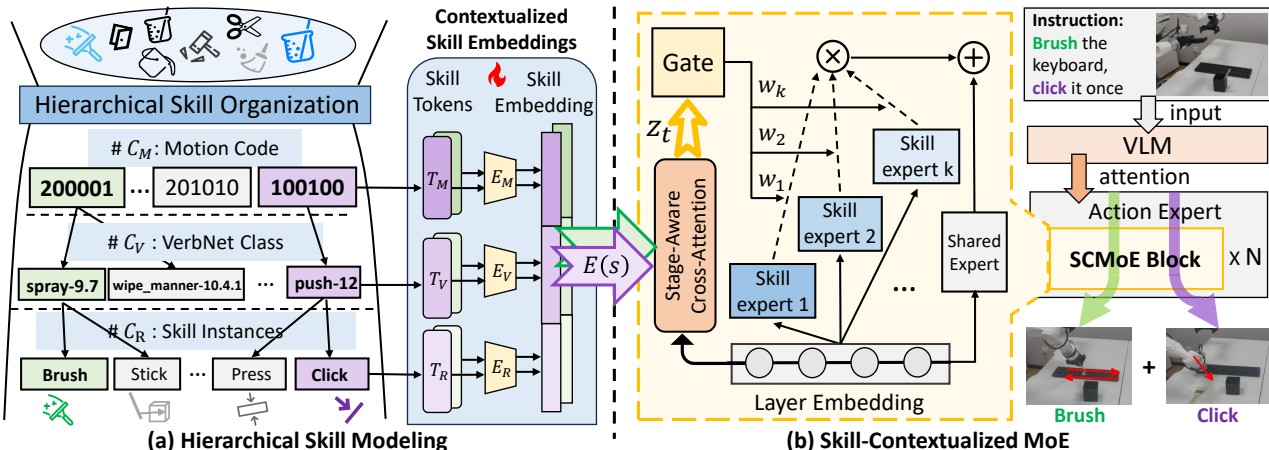

*Figure 2.* The overall pipeline of SkillNet. (a) SkillNet organizes skills with motion categories $c_M$, semantic categories $c_V$, and specific skill realizations $c_R$. Learnable skill embeddings are contextualized by hierarchical skill attributes. (b) Skill-Contextualized MoE use task-relevant skill embeddings as soft MoE routing condition to formulate similar parametric space on similar skills.

## 2.2. Skill-Centric Methods

Skill-centric methods treat various tasks as compositions of repeatable and atomic skills, which provide a possible pathway for compositional generalization in robotic manipulation. Policy code generation approaches abstract atomic skills into code APIs, enabling the generation of executable code that parameterizes predefined skills (Liang et al., 2023; Huang et al., 2023a; Mu et al., 2024; Singh et al., 2023; Huang et al., 2022; 2023b; Xie et al., 2025). These API-based methods are transparent on semantic level, while struggling with scalability of rule-based API implementations. Uni-Skill (Xie et al., 2026) further expands skill implementations with semantic hierarchies, however, it is still restricted by low precision of atomic APIs. Another branch of skill-centric methods do not model skills in an explicit manner (Wan et al., 2024; Liang et al., 2024; Wu et al., 2025). Skills in these methods are discovered from scratch and serve as condition to select different policies for continual learning or compositional tasks. Both branches enhance the capacity of policy models to tackle transitions across different tasks. However, skills are primarily loosely organized, making it difficult for transferring to unseen behaviors. In this work, we propose HSM to organize similarities and differences of skills in a hierarchical way, and explore how to express skill hierarchies in VLA models to achieve compositional generalization.

## 3. Method

### 3.1. Problem Formulation

We consider language-conditioned robotic manipulation where each task is instructed by the language input $\tau$. At each timestep $t$, the observation can be noted as $\mathbf{O}_t = \{\mathbf{q}_t, \{\mathbf{I}_t^1, ...\mathbf{I}_t^N\}, \tau\}$, where $\mathbf{q}_t$ denotes the proprioceptive

state and $\{\mathbf{I}_t^k\}_{k=1}^N$ represents multi-view visual inputs. The objective is to learn a policy $\pi(\mathbf{A}_t|\mathbf{O}_t)$ that generates a sequence of future actions $\{a_t, a_{t+1}, ..., a_{t+H}\}$ with a chunk size of $H$ to complete the task instruction $\tau$.

For a task described by free-form natural language instruction $\tau$, the same underlying task may admit multiple linguistic expressions, and different subtasks can be stacked to form new tasks. Such task-level variability poses a significant challenge for generalization. In contrast, skills capture behavioral commonalities shared across different tasks: they are repeatable and re-combinable primitives. By decomposing tasks into atomic skills, the exponential complexity of the task space can be reduced to learning a set of reusable behavior patterns, which forms a fundamental basis for generalization in compositional tasks.

Meanwhile, skills are not isolated entities. Relevant skills naturally share similar attributes, while such similarities and differences manifest at different levels of granularity. For instance, *pick*, *push*, and *pull* all involve sustained physical contact with the object and thus share a similar contact type. However, *push* and *pull* further share a directional force applied along the contact surface, whereas *pick* differs in its lifting-oriented objective. These observations motivate a hierarchical organization of skill attributes (e.g., contact type, directional constraints), which captures skill relationships at multiple levels of abstraction. Formally, let a skill be denoted as $s$, and let $c(s) = \{c^{(1)}(s), \ldots, c^{(L)}(s)\}$ represent the set of attributes associated with $s$. We introduce a skill hierarchy to organize these attributes across different levels of abstraction, and denote by $H(s)$ the hierarchical organization of attributes for skill $s$.

To exploit transferable attributes in $H(s)$, the robotic policy $\pi(\mathbf{A}_t|\mathbf{O}_t)$ is endowed with an interface to access the

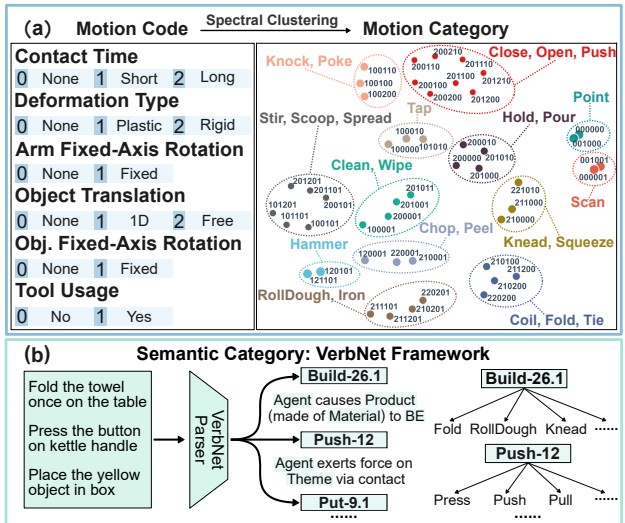

*Figure 3.* An overview of the construction process for motion and semantic categories. (a) Motion code and its clusters with spectral clustering, which formulate the motion categories. (b) The VerbNet Framework, where each verb phrase is converted to a semantically organized category with the verbnet parser.

hierarchy, allowing $\mathrm{H}(s)$ to function as an intermediate constraint that guides the learning process. Rather than learning separate policies $\pi_s$ and $\pi_{s'}$ for different skills $s$ and $s'$, we condition the robotic policy on hierarchical skill attributes. Specifically, we learn a unified skill-conditioned policy

$$\pi(\mathbf{A}_t \mid \mathbf{O}_t, \mathrm{H}(s)). \tag{1}$$

Conditioning on hierarchical skill representations allows shared components among relevant skills to be directly reflected in the policy, and in our implementation, this is manifested as direct sharing and reuse of policy parameters. The construction of a comprehensive skill hierarchy and the design of a skill-conditioned policy correspond to the two core components of SkillNet: the Hierarchical Skill Modeling and the Skill-Contextualized Mixture-of-Experts.

### 3.2. Hierarchical Skill Modeling

As shown in Figure 2 (a), SkillNet organizes diverse skill instances in a three-layer abstraction. Each skill $s$ is associated with a hierarchical decomposition $\mathrm{H}(s) := c_M \rightarrow c_V \rightarrow c_R$, where $c_M$ denotes motion categories capturing mechanical properties characterized by motion codes (Paulius et al., 2020), $c_V$ represents semantic categories summarized using VerbNet classes (Schuler, 2005), and $c_R$ corresponds to the most fine-grained skill realizations (e.g., *push*, *pour*).

We first introduce the motion category $c_M$. As depicted in Figure 3 (a), each motion code in our taxonomy is a six-digit sequence that captures similarities among skills in terms of their mechanical properties (e.g., the motion code for *"Fold clothes on table"* is *210200*). Each digit in the mo-

tion code corresponds to a specific dimension of mechanical properties (contact time, deformation type, arm rotation, object translation, object rotation, tool usage). These explicit motion codes are directly measurable and reside in a compact, low-dimensional space. In contrast to raw trajectories, which are often noisy and unstructured, motion codes allow us to directly characterize skill similarities based on their underlying mechanical properties, without the need for complex trajectory clustering. Leveraging an automatic data curation pipeline built with the off-the-shelf vision–language model (Singh et al., 2025), we annotate motion codes for over 470K demonstrations from two heterogeneous robotic datasets (Khazatsky et al., 2024; Bu et al., 2025). Statistical analysis reveals that these codes form dense clusters under a weighted Hamming distance. By applying spectral clustering, we further distill these clusters into 13 representative categories, as visualized via t-SNE in Figure 3 (a). Meanwhile, annotated motion codes from video demonstrations provide an offline mapping from verb phrases to $c_M$ in HSM, enabling direct access to motion categories during execution. Details of automatic data curation pipeline are provided in the Appendix B.

Beyond mechanical properties, semantic attributes related to target objects, locations, context, and force dynamics also capture execution-relevant information. These attributes are more fine-grained than motion categories and complement motion representations by modeling detailed variations in trajectories and object interactions. Inspired by ImageNet (Deng et al., 2009), which organizes visual entities using WordNet (Miller, 1995), we adopt VerbNet to hierarchically structure semantic categories $c_V$ in HSM. As illustrated in Figure 3(b), a VerbNet parser maps verb phrases to VerbNet classes along with their associated semantic and thematic roles. From the complete set of 277 VerbNet classes, we further curate 127 classes relevant to robotic manipulation, excluding semantics related to biological or subjective behaviors (e.g., *breathe*).

Based on the motion category $c_M$ and the semantic category $c_V$, bottom level skill realizations $c_R$ are grounded in demonstrations from DROID and AgiBot-World, covering over 200 distinct verbal phrases. To express semantically organized skills with learnable representations, we introduce a standard skill tokenization strategy on top of HSM. As shown in Figure 2 (a), each hierarchical skill attribute $\mathrm{H}(s) = (c_M, c_V, c_R)$ is mapped to corresponding skill tokens $(T_M, T_V, T_R)$ and further encoded into a contextualized skill embedding $\mathbf{Emb}(s)$ through dedicated embedding layers $\mathrm{E}_M, \mathrm{E}_V, \mathrm{E}_R$:

$$\mathbf{Emb}(s) = \mathrm{E}_M(T_M) \oplus \mathrm{E}_V(T_V) \oplus \mathrm{E}_R(T_R), \tag{2}$$

where $\oplus$ denotes vector concatenation. At the execution stage, given an observation $O_t$, the language instruction $\tau$ is first processed by a lightweight VLM to extract verb phrases

$\{v_i\}_{i=1}^T$ represent subtasks. Then HSM identifies a pathway from $c_M$, $c_V$ to $c_R$ for each verb phrase $v_i$, converting it into hierarchical skill attributes $\mathrm{H}(s_i)$. Embeddings $\mathbf{Emb}(s_i)$ built on $\mathrm{H}(s_i)$ are further stacked in temporal order to form a sequence of skill embeddings:

$$\mathbf{E}(s_{1:T}) = \big[\mathbf{Emb}(s_1), \mathbf{Emb}(s_2), \dots, \mathbf{Emb}(s_T)\big], \quad (3)$$

where $T$ denotes the number of skills involved in the task. This sequence $\mathbf{E}(s_{1:T})$ serves as input to downstream policy modules, capturing both the hierarchical structure and the sequential composition of the task-relevant skills.

### 3.3. Skill-Contextualized Mixture-of-Experts

Building upon the hierarchical skill modeling, we propose a skill-contextualized Mixture-of-Experts (SCMoE) that models similarities among policy distributions, with expert compositions regulated by relational structures encoded in skill attributes. In each layer of the network, we expand the original dense feedforward module into a convex mixture of structurally identical feedforward experts:

$$h^{(\ell)} = \sum_{e=1}^{E_\ell} w_e^{(\ell)}\big(\mathbf{E}(s_{1:T}), h^{(\ell-1)}\big) \cdot f_e^{(\ell)}\big(h^{(\ell-1)}\big), \quad (4)$$

where $w_e^{(\ell)}$ models the weights of different experts on the same layer and $f_e^\ell$ models the feedforward process for each activated expert. As we formally prove in the Appendix A, this mixture formulation enables the network to capture correlations among policy distributions corresponding to skills with similar attributes, effectively constraining the learned policies within shared subspaces for related skills.

Rather than explicitly injecting a single skill embedding $\mathbf{Emb}(s_t)$ at each timestep, SCMoE employs a cross-attention mechanism between the intermediate representation $h^{(\ell-1)}$ and the sequential skill embeddings $\mathbf{E}(\mathbf{s}_{1:T})$ corresponding to the entire task. This design enables the model to softly attend to different skill stages and smoothly transition across task phases without requiring hard skill boundaries. Formally, the merged skill representation $\mathbf{z}^{(\ell)}$ at layer $\ell$ is computed as:

$$\mathrm{Attn}\Big(\phi_q^{(\ell)}\big(h^{(\ell-1)}\big),\ \phi_k\big(\mathbf{E}(\mathbf{s}_{1:T})\big),\ \phi_v\big(\mathbf{E}(\mathbf{s}_{1:T})\big)\Big). \quad (5)$$

Layers of an MLP followed by a softmax function are applied to $\mathbf{z}^{(\ell)}$ to produce the final expert weights:

$$\mathbf{w}^{(\ell)}\big(\mathbf{E}(s_{1:T}), h^{(\ell-1)}\big) = \mathrm{SoftMax}\big(\mathrm{MLP}^{(\ell)}(\mathbf{z}^{(\ell)})\big), \quad (6)$$

where $w_e^{(\ell)}$ denotes the $e$-th entry of $\mathbf{w}^{(\ell)}$. This modeling of implicit skill composition and transition over task executions allows the network to softly integrate and transition between skills, supporting coherent execution across multi-stage behaviors without relying on costly per-timestep skill

annotations. In practice, we introduce a shared expert which is always activated to stabilize the training process (Yang et al., 2025). Meanwhile, the other experts are sparsely activated with the highest ranking for calculation (only top-K experts are considered), thereby increasing computational efficiency and realizing identical parametric pathways with different expert compositions.

The SCMoE expansion is agnostic to the underlying dense model, making it compatible with a variety of VLA architectures. In practice, we adopt the classical instantiation of the flow-matching model $\pi_{0.5}$ (Black et al., 2025) as the implementation backbone. Flow-matching methods can be decomposed to the VLM trained with auto-regressive process and an expert that iteratively generates actions via the denoising process. We apply SCMoE to the action expert, initializing each skill-contextualized expert with the weights of the dense model to retain the knowledge acquired during pretraining. The flow matching loss $\mathcal{L}_{\mathcal{FM}}$ is defined as:

$$\mathcal{L}_{\mathcal{FM}} = \mathbb{E}\big[||\epsilon - A_t - v_\theta(A_t^\tau, O_t | \mathbf{E}(s_{1:T}))||^2\big], \quad (7)$$

where $\epsilon \sim \mathcal{N}(0, 1)$, $A_t^\tau = (1 - \tau)A_t + \tau\epsilon$ represents the noisy action at the timestep $\tau$, and $v_\theta(A_t^\tau, O_t | \mathbf{E}(s_{1:T}))$ is the learned velocity field that predicts the denoising direction and constrained by the skill embeddings $\mathbf{E}(s_{1:T})$.

To prevent the expert selection strategy from collapsing to a limited set of expert combinations, we introduce a router balance loss to promote diverse and balanced usage of skill-contextualized parametric pathways:

$$\mathcal{L}_{rb} = K \cdot \sum_{i=1}^K f_i P_i, \quad (8)$$

where $f_i$ represents the fraction of tokens for which expert i is selected in the top-K routing, and $P_i$ is the average gating probability for expert i across all tokens before top-k selection. The final training objective can be formulated as: $\mathcal{L}_{SC} = \mathcal{L}_{FM} + \lambda_{rb} \cdot \mathcal{L}_{rb}$, where $\lambda_{rb}$ is the weighting coefficient of the router balance loss. Analysis on computational efficiency is provided in the Appendix F.

## 4. Experiments

To comprehensively evaluate how SkillNet supports compositional generalization in vision-language action models, we conduct three groups of experiments, examining (1) **zero-shot generalization to unseen task compositions**, (2) **few-shot transfer to unseen behaviors**, and (3) effects on **in-domain performance**. All experiments are implemented on both simulators (Liu et al., 2024; Nasiriany et al., 2024; Chen et al., 2025) and real-world environments. We deploy our real-world experiments on a dual-arm AgiBot G1 robot, with parallel grippers and three different camera views (left

*Table 1.* Zero-shot success rate (%) of baseline VLAs and SkillNet on LIBERO-Skill. Task IDs denote skill compositions and each number represents an identical skill (e.g., *"74"* means "turn off stove then close drawer"). Details refer to the main text.

| METHOD | 0101 | 301 | 401 | 014 | 46 | 32 | 501 | 601 | 74 | AVG. |
|---|---|---|---|---|---|---|---|---|---|---|
| OPENVLA-OFT (KIM ET AL., 2025) | 0.0 | 0.0 | 0.0 | 0.0 | 0.0 | 0.0 | 0.0 | 0.0 | 0.0 | 0.0 |
| OPENVLA (KIM ET AL., 2024) | 0.0 | 0.0 | 18.0 | 0.0 | 0.0 | 0.0 | 0.0 | 0.0 | 0.0 | 2.0 |
| $\pi_0$ (BLACK ET AL., 2024) | 0.0 | 0.0 | 22.0 | 4.0 | 0.0 | 0.0 | 0.0 | 4.0 | 0.0 | 3.3 |
| GR00T-N1.6 (BJORCK ET AL., 2025) | 2.0 | **96.0** | 2.0 | 2.0 | 8.0 | 0.0 | 0.0 | 0.0 | 0.0 | 12.2 |
| $\pi_{0.5}$ (BLACK ET AL., 2025) | 58.0 | 16.0 | 88.0 | 22.0 | 20.0 | 14.0 | 8.0 | **60.0** | 0.0 | 31.8 |
| VANILLA MoE | **74.0** | 26.0 | 90.0 | 38.0 | 2.0 | 10.0 | 14.0 | 44.0 | 0.0 | 33.1 |
| **SKILLNET** | **74.0** | 56.0 | **92.0** | **60.0** | **28.0** | **24.0** | **20.0** | 56.0 | **20.0** | **47.8** |

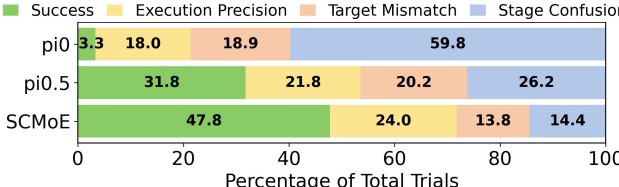

*Figure 4.* Failure modes of different models on LIBERO-Skill.

*Table 2.* Zero-shot success rate on real-world compositional tasks.

| TASK | $\pi_{0.5}$ | VANILLA MoE | SKILLNET |
|---|---|---|---|
| DRAG + CLICK | 70.0 | 85.0 | **90.0** |
| SWEEP + SHAKE | 5.0 | 25.0 | **65.0** |
| BRUSH + DRAG | 25.0 | 20.0 | **45.0** |
| WIPE + CLICK | **15.0** | 5.0 | **15.0** |
| OPEN + CLOSE | 65.0 | 60.0 | **80.0** |
| DRAG + SHAKE | 75.0 | 70.0 | **80.0** |
| AVG | 42.5 | 44.2 | **62.5** |

arm, right arm, and head). Demonstrations and code are provided in our project page[1].

### 4.1. Generalize to Unseen Task Compositions

First, we evaluate the zero-shot generalization capability of SkillNet on unseen task compositions. Existing simulation benchmarks primarily focus on variations of positions, lighting conditions, and target objects, while rarely considering generalization over different task compositions. To this end, we propose a new task suite on LIBERO (Liu et al., 2024), named **LIBERO-Skill**. LIBERO-Skill contains 9 different tasks, and each task is composed of different skill sequences. Tasks in LIBERO-Skill are specified by sequences of skill identifiers, where different numeric labels correspond to distinct skills (0: pick, 1: place, 2: stack, 3: open-prismatic, 4: close-prismatic, 5: open-revolute, 6: close-revolute, 7: turn). These identifiers are concatenated to form task-level skill sequences. Each skill sequence in LIBERO-Skill is unseen during pre-training on LIBERO-90, while the individual skills themselves are present in the training data. This design enables us to consider generalization on task compositions independently, without conflicting with difficulties on unseen behaviors or objects. We provide details of the LIBERO-Skill task suite in the Appendix D.2.

We report the average success rate with 50 trials for each task on LIBERO-Skill. The compared baselines contain Open-VLA (Kim et al., 2024), OpenVLA-oft (Kim et al., 2025), and the state-of-the-art flow-matching VLAs ($\pi_0$ (Black et al., 2024), $\pi_{0.5}$ (Black et al., 2025), GR00T-N1.6 (Bjorck et al., 2025)). We also implement the Vanilla MoE on $\pi_{0.5}$

---

[1]https://github.com/VIPL-VSU/SkillNet

as a comparison with SCMoE in SkillNet. All baselines and SkillNet are trained on LIBERO-90 with 50 demonstrations each task, and evaluated in a zero-shot manner on LIBERO-Skill. As shown in Table 1, we find that earlier methods trained exclusively on robotic data (OpenVLA, OpenVLA-oft, $\pi_0$) struggle to generalize in a zero-shot manner to unseen task compositions. Despite recent VLAs (GR00T-N1.6, $\pi_{0.5}$) exhibiting improved embodied reasoning capabilities and achieving non-zero results, their overall performance remains relatively low and uneven. Compared with the state-of-the-art baseline $\pi_{0.5}$, SkillNet shows an improvement of 16.0%. This improvement stems from modeling repeatable skills and transition stages, enabling effective reuse across compositional behaviors. We further observe that replacing SCMoE with a vanilla MoE on the same backbone does not yield comparable improvements, suggesting that the gains in zero-shot generalization mainly stem from structured skill modeling and reuse rather than the MoE formulation itself.

We further analyze the failure modes of different models on the LIBERO-Skill suite. Failures on long-horizon tasks can be categorized into three types: Stage Confusion, Target Mismatch, and Execution Precision. Stage Confusion arises from stuck or random actions during skill transitions. Target Mismatch occurs when the target object is wrong. Execution Precision refers to inaccuracies in execution. As shown in Figure 4, SkillNet exhibits a significant reduction in Stage Confusion errors compared with the baseline $\pi_{0.5}$, which can be attributed to its improved modeling of repetitive skills and transition stages. A similar 6.4% reduction is observed for Target Mismatch, indicating better target perception across diverse task compositions. Execution Pre-

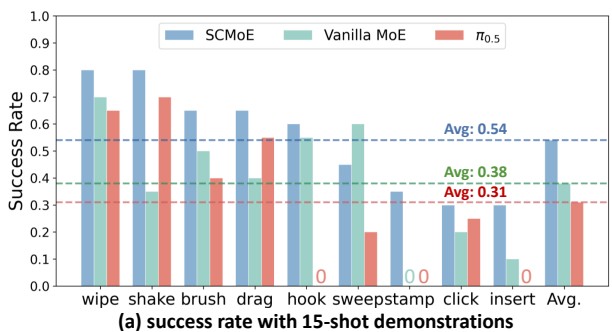 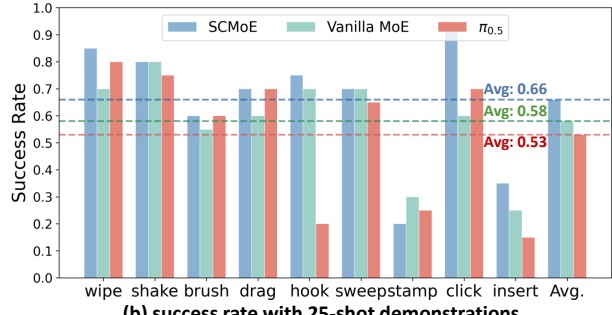

*Figure 5.* Success rate (%) of baselines and SkillNet on 9 transfer skills under different few-shot settings.

cision errors, by contrast, only occur in the absence of the first two types of failures; as a result, SkillNet shows a slight increase in this category compared to other models.

In real-world environments, we first pre-trained SkillNet and the baselines ($\pi_{0.5}$ and vanilla MoE) on 21 tasks with different skill compositions. We then designed six different skill combinations that are unseen in the training set to evaluate the zero-shot generalization capabilities, averaged over 20 trials per task. As shown in Table 2, SkillNet shows a consistent improvement over the baselines $\pi_{0.5}$ and vanilla MoE by 20.0% and 18.3%. During execution, SkillNet exhibits noticeably fewer failures caused by skill transition issues and attains higher execution precision on primitive skills observed during training. These observations are consistent with the trends found in simulation experiments. Details can be found in the Appendix E.3.

### 4.2. Transfer to Unseen Behaviors

Furthermore, we evaluate the few-shot transfer ability of SkillNet to *unseen behaviors*. Tasks are partitioned into two sets based on their skill compositions: a pre-training set and a transfer set. The pre-training set is curated to cover a broad spectrum of motion patterns, establishing a foundation for transfer to skills sharing similar mechanical properties. In contrast, the transfer set consists of tasks that share mechanical similarities with the pre-trained ones but introduce novel execution dynamics or targets. This ensures that the evaluation targets functional adaptation rather than superficial variations, such as simple linguistic synonyms (e.g., *place* vs. *put*). Models are first trained on the pre-training set to internalize a diverse range of motion patterns. Subsequently, they are provided with few-shot demonstrations to adapt these learned behaviors to the transfer set.

For real-world evaluation, we curated a pre-training suite comprising 12 distinct tasks that encompass 8 fundamental motion patterns. The transfer set consists of 9 skills that share mechanical properties with the pre-training tasks but differ in execution-critical dimensions, such as target trajectories and force dynamics, rather than mere semantic proximity. For example, despite pretrained skill *brush* and

*Table 3.* Success rate (%) on LIBERO under in-domain settings.

| METHOD | SPATIAL | OBJECT | GOAL | LONG | AVG. |
|---|---|---|---|---|---|
| OPENVLA | 84.7 | 88.4 | 79.2 | 53.7 | 76.5 |
| $\pi_0$ | 96.4 | 98.8 | 95.8 | 85.2 | 94.2 |
| OPENVLA-OFT | 97.7 | 98.0 | 96.1 | 95.3 | 96.8 |
| GR00T-N1.6 | 97.7 | 97.5 | 98.5 | 94.4 | 97.0 |
| ATOMICVLA | 98.8 | 98.8 | 97.2 | 96.2 | 97.8 |
| $\pi_{0.5}$ | 99.4 | 97.8 | 98.2 | 96.6 | 98.0 |
| **SKILLNET** | **99.6** | **99.2** | **99.0** | **98.4** | **99.1** |

evaluated skill *erase* both involve contact-rich tool manipulation, they differ on semantic objectives and force dynamics. All evaluated models ($\pi_{0.5}$, $\pi_{0.5}$ with vanilla MoE implementation, and SkillNet) were pre-trained with same hyperparameters on the pre-training tasks. Subsequently, these models were fine-tuned on transfer tasks using few-shot demonstrations (15 and 25 shots). Each task was evaluated over 20 trials with randomized scene layouts, and we report the average success rate. Detailed descriptions of all skills are provided in the Appendix E.2.

As shown in Figure 5, the performance of SkillNet shows a significant improvement compared with baselines ($\pi_{0.5}$, $\pi_{0.5}$ with vanilla MoE) under a few-shot transfer setting. On the 9 evaluation tasks, SkillNet achieves a 23.9% improvement in average performance over $\pi_{0.5}$ in the 15-shot regime. Notably, 15-shot performance of SkillNet approaches the performance of baselines on with 25 demonstrations, demonstrating superior sample efficiency. In addition, we observe that SkillNet better captures the onset and termination of behaviors, leading to fewer stalls and appropriate termination in cyclic skills such as *brush* or *sweep*. This improvement primarily stems from hierarchical modeling of skill attributes and skill-contextualized parametric pathways in SkillNet. Attribute sharing across skills is realized via shared expert compositions, enabling structured parameter reuse and compositional generalization to unseen behaviors.

The simulation experiments are conducted on RoboTwin-2.0 (Chen et al., 2025), with models pretrained on 15 tasks and evaluated on 15 different tasks under few-shot settings. Despite limited skill diversity and low overlap between

*Table 4.* Success rate (%) on long-horizon real-world tasks under in-domain settings.

| METHOD | BRU+CLK | OP+PP+CLS | CUT+STP | PP+STR | PP+DRG | PP+SHK | SWP+WIP | AVG. |
|---|---|---|---|---|---|---|---|---|
| $\pi_{0.5}$ | 15.0 | 65.0 | 10.0 | 35.0 | **90.0** | **80.0** | 35.0 | 47.1 |
| VANILLA MOE | 40.0 | 60.0 | **15.0** | 25.0 | 85.0 | 70.0 | **55.0** | 50.0 |
| **SKILLNET** | **45.0** | **85.0** | **15.0** | **40.0** | **90.0** | 65.0 | 50.0 | **55.7** |

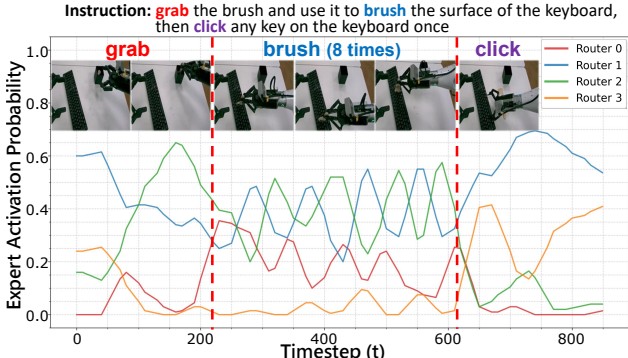

*Figure 6.* Router transitions on a long-term real-world task.

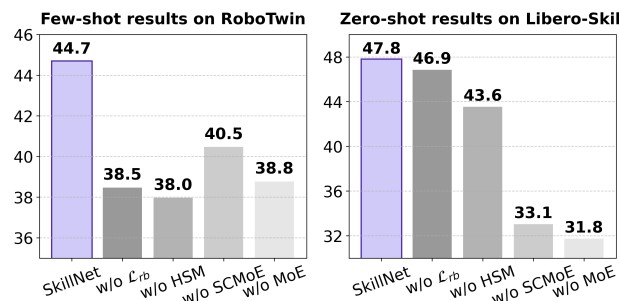

*Figure 7.* Success rates (%) of different ablation models under zero-shot and few-shot settings.

skills, SkillNet still outperforms $\pi_{0.5}$ by 5.9% under the 15-shot setting. This further validates the robustness of SkillNet in few-shot transfer scenarios. Detailed results on RoboTwin are provided in the Appendix D.1.

## 4.3. In-Domain Performance

Finally, we validated that the framework of SkillNet is also beneficial for the in-domain performance of VLA models. The simulation experiments are implemented on two well-known benchmarks LIBERO (Liu et al., 2024) and Robo-Casa (Nasiriany et al., 2024). The baselines contain Open-VLA (Kim et al., 2024), OpenVLA-oft (Kim et al., 2025), and the flow matching models (Black et al., 2024; 2025; Bjorck et al., 2025; Zhang et al., 2026). All baselines are trained with the same demonstration data in a multi-task setting. For LIBERO, we adopt the standard four suite setting, where each suite contains 10 different tasks. Each task is provided with 50 demonstrations and the average success rate is reported on 50 trials. As shown in Table 3, SkillNet achieves an average success rate of **99.1%** on LIBERO, surpassing all other baselines. For RoboCasa, we report the success rate on 24 tasks with the Franka Panda arm. Each task is provided with 100 demonstrations and evaluated on 50 trials. The performance of SkillNet on RoboCasa also surpasses the baseline $\pi_{0.5}$ and its vanilla MoE implementation by 1.7%. Detailed performance on RoboCasa is provided in the Appendix D.3.

We further evaluated SkillNet on seven long-horizon real-world tasks under in-domain settings. Each task is provided with 50 demonstrations and evaluated with 20 trials. As shown in Table 4, SkillNet outperforms the vanilla MoE implementation on $\pi_{0.5}$ by 5.7%, which is consistent with

results on simulators. This suggests that the benefits of Skill-Net are not limited to out-of-distribution generalization, but may also extend to standard multi-task learning, where skill-aware routing helps organize shared parameters according to related behaviors and skill transitions. Details of real-world in-domain tasks are provided in the Appendix E.4.

## 4.4. Routing Analysis and Interpretability

To further analyze how similarities on hierarchical skill attributes influence the routing strategy, we did representational similarity analysis (RSA) between skill attributes and the expert activation probabilities at a given expert layer. For SCMoE, the resulting correlation coefficient reaches $\rho = 0.48$ across 12 real-world skills, suggesting that inter-skill similarities are substantially reflected in the routing behavior during execution. In contrast, the Vanilla MoE exhibits a near-zero correlation, indicating a more fragmented and task-specific routing pattern. This alignment between skill attributes and routing behavior suggests that SCMoE fosters a shared parametric space for relevant skills, which serves as a foundation for positive transfer on unseen behaviors. Details of RSA are provided in the Appendix C.

We also observe interpretable relationships between expert activations and semantic skills. As shown in Figure 6, different semantic skills (e.g., grab, brush, and click) are associated with distinct expert compositions rather than individual experts. Moreover, transitions between skill phases are accompanied by corresponding shifts in the activated expert compositions, indicating that the router dynamically adapts expert usage according to the current skill being executed.

In addition to this skill-level compositional structure, we examine whether individual experts exhibit preferences over lower-level motion patterns. We find that such preferences

also emerge over fundamental mechanical properties represented by motion codes. For example, in the first expert layer, one expert is more frequently activated for skills involving short-term forceful contact (e.g., click, strike, and stamp), whereas another expert is preferred for skills involving long-term tool-mediated contact (e.g., brush, wipe, sweep, and scoop). A third expert tends to specialize in repetitive motions such as brush, sweep, and strike.

## 4.5. Ablations

To evaluate the effectiveness of modules in SkillNet, we did ablation studies on model architecture, data structure, and training strategies, corresponding to the design of SCMoE, HSM, and router balance loss. These ablations are conducted on both few-shot and zero-shot settings. In addition, since HSM relies on automatically curated skill attributes, we include a sensitivity analysis on LIBERO-Skill to examine the robustness of SkillNet to noisy skill tags.

**The role of SCMoE.** To validate the effectiveness of SkillNet, we replace SCMoE with a vanilla MoE while keeping all other settings consistent to train our model. As shown in Figure 7, SCMoE plays the most important role on zero-shot generalization. By effectively capturing skill correlations and transitions, it serves as the foundation for compositional generalization capabilities on the network level.

**The role of HSM.** To assess the effectiveness of HSM, we replaced it with a flat skill organization where skill embeddings are no longer contextualized by hierarchical attributes. Instead, each skill is represented as an independent categorical ID. As shown in Figure 7, removing HSM significantly degrades performance in few-shot transfer to unseen behaviors. This suggests that hierarchical skill modeling is pivotal for achieving robust generalization across novel tasks.

**The role of the router balance loss.** We further examine the role of the router balance loss $\mathcal{L}_{rb}$ during training. As shown in Figure 7, the router balance loss plays an important role on few-shot transfer between skills. The success rate degrades by 6.2% when $\mathcal{L}_{rb}$ is removed during training. This stems from improved skill utilization induced by $\mathcal{L}_{rb}$, which prevents routing collapse and maintains richer expert compositions to express hierarchical skill attributes.

**Robustness to noisy skill attributes.** On LIBERO-Skill, we randomly replaced skill tokens with incorrect motion or semantic tags at different probabilities. Even with 20% corrupted skill tokens, SkillNet drops by only 4.0% and still outperforms the $\pi_{0.5}$ baseline by 12%, suggesting that SCMoE is robust to moderate tagging errors. We also analyze the VLM-generated annotations and find that only 7% of cases contain minor deviations in object rigidity annotations, with limited impact on overall performance.

## 5. Conclusion

In this paper, we propose SkillNet, a skill-centric framework that organizes diverse skills with hierarchical skill modeling, and leverages Skill-Contextualized Mixture-of-Experts to express transferable attributes across skills. By enabling relevant skills to share a compositional parametric space and recombine across various tasks, SkillNet facilitates robust compositional generalization in VLAs. Our approach demonstrates significant improvements over state-of-the-art methods in both zero-shot generalization and few-shot adaptation to novel behaviors. These findings suggest that an expressive hierarchy encompassing both motion and semantics provides a powerful inductive bias for policy design. However, the optimal depth of such hierarchies and the potential benefits of even finer-grained skill decompositions remain open questions for further exploration.

## Acknowledgement

This work is partially supported by Beijing Municipal Natural Science Foundation Nos. L257009, L242025, Natural Science Foundation of China under contracts Nos. 62495082, 62461160331, 62476011.

## Impact Statement

This paper presents work whose goal is to advance the field of Machine Learning. There are many potential societal consequences of our work, none of which we feel must be specifically highlighted here.

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

## A. Skill Similarity and Policy Correlation under Top-$K$ MoE

We analyze how skill similarity, characterized by shared attributes, induces correlated policy distributions under a multi-layer Top-$K$ Mixture-of-Experts (MoE) architecture.

**Top-$K$ MoE Routing.** At each MoE layer $\ell \in \{1, \ldots, L\}$, a routing network produces expert logits $r_e^{(\ell)}(C)$ conditioned on the skill attribute representation $C$. The Top-$K$ routing mechanism selects a sparse subset $\mathrm{TopK}^{(\ell)}(C) \subset \{1, \ldots, E_\ell\}$, with $|\mathrm{TopK}^{(\ell)}(C)| = K$, and assigns normalized mixture weights

$$w_e^{(\ell)}(C) \propto \exp(r_e^{(\ell)}(C)) \ \text{ for } e \in \mathrm{TopK}^{(\ell)}(C), \tag{9}$$

while $w_e^{(\ell)}(C) = 0$ otherwise.

**Policy as a Sparse Mixture over Expert Paths.** Stacking $L$ MoE layers, the induced policy can be written as a sparse mixture over expert paths $\mathbf{e} = (e_1, \ldots, e_L)$:

$$\pi_C(a \mid o) = \sum_{\mathbf{e} \in \mathcal{P}(C)} \alpha_{\mathbf{e}}(C) \, \pi_{\mathbf{e}}(a \mid o), \tag{10}$$

where

$$\mathcal{P}(C) = \prod_{\ell=1}^{L} \mathrm{TopK}^{(\ell)}(C), \qquad |\mathcal{P}(C)| = K^L, \tag{11}$$

and the path-level mixture weight factorizes across layers as

$$\alpha_{\mathbf{e}}(C) = \prod_{\ell=1}^{L} w_{e_\ell}^{(\ell)}(C), \qquad \sum_{\mathbf{e}} \alpha_{\mathbf{e}}(C) = 1. \tag{12}$$

**Routing Overlap Induced by Skill Similarity.** Consider two skills with attribute representations $C$ and $C'$. We define the Top-$K$ routing overlap at layer $\ell$ as

$$\mathrm{Overlap}^{(\ell)}(C, C') = \frac{\left| \mathrm{TopK}^{(\ell)}(C) \cap \mathrm{TopK}^{(\ell)}(C') \right|}{K}. \tag{13}$$

Assuming the routing logits $r^{(\ell)}(\cdot)$ are continuous with respect to $C$, skills sharing similar attributes induce similar expert rankings, resulting in a high routing overlap across layers.

**Policy Correlation via Shared Expert Paths.** Let $\alpha_C$ and $\alpha_{C'}$ denote the path-level mixture distributions induced by $C$ and $C'$, respectively. The total variation distance between the resulting policies satisfies

$$\mathrm{TV}(\pi_C(\cdot \mid o), \pi_{C'}(\cdot \mid o)) \ \leq \ \frac{1}{2} \left\| \alpha_C - \alpha_{C'} \right\|_1. \tag{14}$$

Since non-overlapping routing paths contribute all of the $\ell_1$ difference, the above distance is bounded by the fraction of expert paths that are not shared. In particular, if the routing overlap at each layer is lower bounded by $\rho_\ell \in (0, 1]$, we obtain

$$\mathrm{TV}(\pi_C(\cdot \mid o), \pi_{C'}(\cdot \mid o)) \ \leq \ 1 - \prod_{\ell=1}^{L} \rho_\ell. \tag{15}$$

**Tightened Bound via Local Lipschitz Continuity.** To establish a functional relationship between attribute divergence and policy variation, we analyze the smoothness of the weighting function. While Top-$K$ selection introduces discontinuities at decision boundaries, the routing function is *locally Lipschitz* within regions where the Top-$K$ expert set remains constant. Inside such a region, let $\mathbf{w}^{(\ell)}(C) = \mathrm{Softmax}(f^{(\ell)}(C))$, where $f^{(\ell)}$ is an MLP with Lipschitz constant $L_f^{(\ell)}$ (with respect to the $L_\infty$ norm of the output).

Using the property that the Softmax function is Lipschitz continuous with constant 2 mapping from $L_\infty$ to $L_1$ metric space, we have:

$$\|\mathbf{w}^{(\ell)}(\boldsymbol{C}) - \mathbf{w}^{(\ell)}(\boldsymbol{C'})\|_1 \leq 2\|f^{(\ell)}(\boldsymbol{C}) - f^{(\ell)}(\boldsymbol{C'})\|_\infty \leq 2L_f^{(\ell)}\|\boldsymbol{C} - \boldsymbol{C'}\|. \tag{16}$$

Applying the telescoping sum inequality for the $\ell_1$ distance of product distributions, the total variation distance is bounded by the sum of layer-wise variations:

$$\mathrm{TV}(\pi_{\boldsymbol{C}}(\cdot \mid o), \pi_{\boldsymbol{C'}}(\cdot \mid o)) \leq \frac{1}{2}\sum_{\ell=1}^{L}\|\mathbf{w}^{(\ell)}(\boldsymbol{C}) - \mathbf{w}^{(\ell)}(\boldsymbol{C'})\|_1 \leq \left(\sum_{\ell=1}^{L} L_f^{(\ell)}\right)\|\boldsymbol{C} - \boldsymbol{C'}\|. \tag{17}$$

**Implication.** This bound demonstrates that for proximal skills (where the expert selection is stable), the policy divergence is linearly constrained by the attribute divergence. The MoE architecture thus acts as a structured bottleneck, ensuring that smoothness in the attribute space translates to correlated policy distributions.

## B. Automatic Data Curation on Skill Attributes

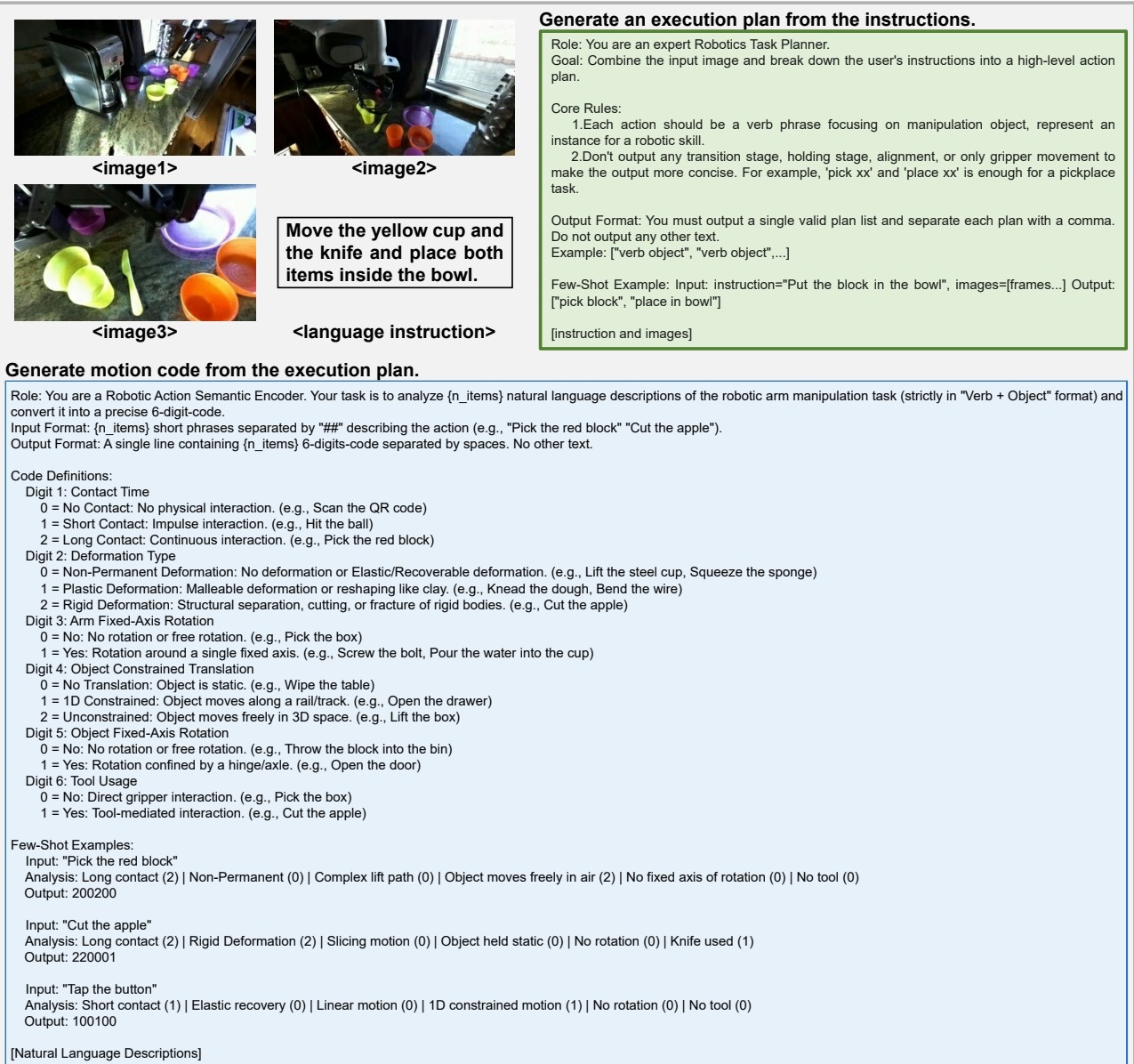

*Figure 8.* Example prompt for plan and motion code generation. We use gpt-5-mini (Singh et al., 2025) as the light-weight VLM to generate verb phrases and motion codes automatically.

## C. RSA analysis

To further quantify the structural correspondence between the pre-defined skill attributes and the emergent expert routing strategies, we employ Representational Similarity Analysis (RSA). This analytical method enables us to evaluate the extent to which the topological structure of the semantic space of skill attributes is preserved within the latent policy space of expert routing probabilities. Specifically, the analysis involves constructing and correlating two Representational Dissimilarity Matrices (RDMs) for the set of $N$ skills, denoted as $\mathcal{S} = \{s_1, s_2, \ldots, s_N\}$:

**The Model RDM ($RDM_{\text{model}}$)** This matrix represents the theoretical semantic distances between skills derived from their motion codes. Given the hierarchical nature of motion code definitions, standard Euclidean metrics are ill-suited. Instead, for any pair of skills $(s_i, s_j)$ with motion code vectors $\mathbf{c}_i$ and $\mathbf{c}_j$, we utilize the *Weighted Hamming Distance*:

$$D_{\text{model}}(i, j) = \frac{\sum_k w_k \cdot \mathbb{I}(c_{i,k} \neq c_{j,k})}{\sum_k w_k} \tag{18}$$

Here, $\mathbb{I}(\cdot)$ denotes the indicator function, and $w_k$ represents the importance weight assigned to the $k$-th attribute dimension. This weighting mechanism allows us to embed domain knowledge into the dissimilarity measure, prioritizing fundamental semantic features over subtle variations.

**The Neural RDM ($RDM_{\text{neural}}$)** This matrix characterizes the divergence in physical control strategies as perceived by the MoE router. Since the gating network outputs sparse probability distributions over experts, commonly used metrics like Euclidean distance or KL divergence may suffer from numerical instability or insensitivity to zero-probability events. We therefore employ the *Hellinger Distance*, a metric defined on the statistical manifold that naturally handles sparse distributions and satisfies the triangle inequality:

$$D_{\text{neural}}(i, j) = \frac{1}{\sqrt{2}} \sqrt{\sum_{m=1}^{M} \left( \sqrt{p_{i,m}} - \sqrt{p_{j,m}} \right)^2} \tag{19}$$

where $\mathbf{p}_i$ and $\mathbf{p}_j$ are the router probability distributions for skills $s_i$ and $s_j$, and $M$ is the total number of experts.

**Statistical Alignment** To quantify the alignment between the semantic intention and the emergent routing strategy, we compute the Spearman's rank correlation coefficient ($\rho$) between the vectorized upper triangular elements of the two RDMs:

$$\rho = \frac{\text{cov}(\text{rank}(\mathbf{v}_{\text{model}}), \text{rank}(\mathbf{v}_{\text{neural}}))}{\sigma_{\text{rank}(\mathbf{v}_{\text{model}})} \sigma_{\text{rank}(\mathbf{v}_{\text{neural}})}} \tag{20}$$

where $\mathbf{v}_{\text{model}}$ and $\mathbf{v}_{\text{neural}}$ are the vectorized upper triangular parts of $RDM_{\text{model}}$ and $RDM_{\text{neural}}$, respectively. A statistically significant positive $\rho$ implies that the router has spontaneously learned a specialized structure reflecting the underlying skill attributes. To assess statistical significance effectively, we perform a non-parametric permutation test with 2,000 iterations, where the labels of the Neural RDM are randomly shuffled to generate a null distribution of correlation coefficients. To validate the effectiveness of our proposed method, we conducted a comparative RSA on a subset of 12 skills against Vanilla MoE. The Vanilla MoE yields a non-significant correlation ($\rho = 0.08$, $p = 0.285$), suggesting that its router utilizes a disjointed or purely low-level feature-driven strategy that ignores semantic topology. In contrast, our Skill-Contextualized MoE achieves a robust and significant alignment ($\rho = 0.48$, $p = 0.001$). This implies that our method does not just learn to solve tasks, but learns a structured policy that mirrors the human-defined semantic hierarchy.

## D. Details on Simulation Tasks

### D.1. Transfer on RoboTwin

RoboTwin 2.0 serves as a scalable simulation framework engineered for high-fidelity, large-scale data synthesis. At its core is the RoboTwin-OD asset library, featuring 731 diverse objects across 147 categories, each enriched with fine-grained semantic and manipulation annotations. To enhance policy robustness and narrow the sim-to-real gap, the framework implements structured domain randomization across five critical dimensions: object clutter, illumination, background textures, tabletop heights, and linguistic instructions.

The RoboTwin 2.0 benchmark consist of 50 tasks encompassing 12 distinct manipulation skills. To construct the pretraining dataset, we curated 15 heterogeneous tasks that span a broad spectrum of fundamental motion categories. This dataset comprises 50 expert trajectories per task, providing a rich foundation for generalizable skill learning. To assess the framework's extrapolative capabilities, we evaluated the model on 15 unseen tasks. These evaluation tasks include skills that are congruent with the pretraining set (e.g., pick, lift, place), as well as entirely novel skill categories (e.g., hang, stamp). The pretrained model was fine-tuned on 50 single-task trajectories to rigorously measure its few-shot adaptation performance. The experimental parameters are detailed in Table 16, while the specifics of the pre-trained and evaluation skills are provided in Tables 6 and 7, respectively.

*Table 5.* Key parameters for the RoboTwin transfer experiment. Transfer tasks are trained separately on RoboTwin.

| Parameter | Value | |
|---|---|---|
| | **Pretrain** | **Transfer** |
| *Training Config* | | |
| Batch size | 32 | 32 |
| Total training steps | **20,000** | **1,000** |
| *Optimization* | | |
| Peak learning rate | $2.5 \times 10^{-5}$ | $2.5 \times 10^{-5}$ |
| Optimizer | AdamW | AdamW |
| $\beta_1, \beta_2$ | 0.9, 0.95 | 0.9, 0.95 |
| EMA decay | 0.99 | 0.99 |
| *Model Architecture* | | |
| Action horizon | 10 | 10 |
| Number of experts | 4 | 4 |
| Top-k selection | 1 | 1 |
| Dimension of skill embeddings | 64 | 64 |

## Pre-train Skills

*Table 6.* The task setting of pre-trained skills on RoboTwin-2.0.

| Task | Skill | Task Instruction | Task Setting |
|---|---|---|---|
| Adjust Bottle | pick | Pick up the bottle on the table headup with the correct arm. |  |
| Beat Block Hammer | pick strike | There is a hammer and a block on the table, use the arm to <grab the hammer> and <beat the block>. |  |
| Click Alarmclock | press | Click the alarm clock's center of the top side button on the table. |  |
| Click Bell | press | Click the <bell's top center> on the table. |  |
| Grab Roller | pick pick | Use both arms to grab the roller on the table. |  |

*Continued on next page*

*Table 6.* The task setting of pre-trained skills on RoboTwin-2.0 (Continued).

| Task | Skill | Task Instruction | Task Setting |
|------|-------|------------------|--------------|
| Handover Block | pick pass place | Use the left arm to grasp the red block on the table, handover it to the right arm and place it on the blue pad. |  |
| Lift Pot | pick pick lift | Use BOTH!!! arms to lift the pot. |  |
| Move Can Pot | pick place | There is a can and a pot on the table, use one arm to <pick up the can> and <move it to beside the pot>. |  |
| Move Playingcard Away | pick place | Use the arm to <pick up the playing card> and <move it away from the table>.For example, if the playing card is on the outward side of the table, you should move it further outward side of the table. |  |
| Open Microwave | open | Use one arm to open the microwave. |  |
| Place Burger Fries | pick pick place place | Use dual arm to pick the hamburg and frenchfries and put them onto the tray. |  |
| Place Object Basket | pick place pick place | Use one arm to grab the target object and put it in the basket, then use the other arm to grab the basket, and finally move the basket slightly away. |  |
| Rotate QRcode | pick rotate | Use arm to catch the qrcode board on the table, pick it up and rotate to let the qrcode face towards you. |  |
| Shake Bottle Horizontally | pick shake place | Shake the bottle horizontally with proper arm. |  |
| Stack Blocks Two | pick place pick place | There are two blocks on the table, the color of the blocks is <red, green>, <move the blocks to the center of the table>, and <stack the green block on the red block>. |  |

**Transfer Skills**

*Table 7.* The task setting of transfer skills on RoboTwin-2.0.

| Task | Skill | Task Instruction | Task Setting |
|---|---|---|---|
| Blocks Ranking Size | pick place pick place pick place | There are three blocks on the table, the color of the blocks is random, move the blocks to the center of the table, and arrange them from largest to smallest, from left to right. |  |
| Hanging Mug | pick rotate place pick hang | Use left arm to pick the mug on the table, rotate the mug and put the mug down in the middle of the table, use the right arm to pick the mug and hang it onto the rack. |  |
| Move Pillbottle Pad | pick place | Use one arm to pick the pillbottle and place it onto the pad. |  |
| Open Laptop | pick lift | Use one arm to open the laptop. |  |
| Place A2B Left | pick place | Use appropriate arm to place object A on the left of object B. |  |
| Place Bread Basket | pick pick place | If there is one bread on the table, use one arm to grab the bread and put it in the basket, if there are two breads on the table, use two arms to simultaneously!!! grab up two breads and put them in the basket. |  |
| Place Break Skillet | pick place | If there is one bread on the table, use one arm to grab the bread and put it into the skillet. |  |
| Place Cans Plasticbox | pick place pick place | Use dual arm to pick and place cans into plasticbox. |  |
| Place Fan | pick place | Grab the fan and place it on a colored mat, <make sure the fan is facing the robot!(THIS MUST BE REFERRED TO)>. |  |
| Press Stapler | press | Use one arm to press the stapler. |  |

*Table 7.* The task setting of transfer skills on RoboTwin-2.0 (Continued).

| Task | Skill | Task Instruction | Task Setting |
|------|-------|------------------|--------------|
| Scan Object | pick pick scan | Use one arm to pick the scanner and use the other arm to pick the object, and use the scanner to scan the object. |  |
| Shake Bottle | pick shake place | Shake the bottle with proper arm. |  |
| Stack Blocks Three | pick place pick place pick place pick place pick place | There are three blocks on the table, the color of the blocks is <red, green and blue>, <move the blocks to the center of the table>, and <stack the blue block on the green block, and the green block on the red block>. |  |
| Stack Bowls Two | pick place | Stack the two bowls on top of each other |  |
| Stamp Seal | pick stamp | Grab the stamp and stamp onto the specific color mat. |  |

## Performance

*Table 8.* Evaluation results of SkillNet, baselines, and ablation studies on RoboTwin Transfer.

| Task | Ablation(Motion Code) | Ablation(Balance Loss) | MoE | $\pi_0$ | $\pi_{0.5}$ | SkillNet |
|------|------|------|------|------|------|------|
| **Blocks Ranking Size** | 2.0 | 2.0 | 0.0 | 0.0 | 0.0 | **4.0** |
| **Hanging Mug** | 0.0 | **4.0** | 0.0 | 2.0 | **4.0** | 0.0 |
| **Move Pillbottle Pad** | 74.0 | 58.0 | 84.0 | 22.0 | 66.0 | **86.0** |
| **Open Laptop** | 44.0 | **62.0** | 52.0 | 48.0 | 50.0 | 52.0 |
| **Place A2B Left** | **24.0** | 14.0 | 14.0 | 4.0 | 14.0 | 16.0 |
| **Place Bread Basket** | 50.0 | **70.0** | 64.0 | 24.0 | 60.0 | 66.0 |
| **Place Bread Skillet** | 54.0 | 40.0 | 48.0 | 14.0 | 38.0 | **60.0** |
| **Place Cans Plasticbox** | 54.0 | 52.0 | 50.0 | 24.0 | 58.0 | **64.0** |
| **Place Fan** | **26.0** | 16.0 | 22.0 | 8.0 | 16.0 | 22.0 |
| **Press Stapler** | 58.0 | 18.0 | 52.0 | **62.0** | 48.0 | **62.0** |
| **Scan Object** | 8.0 | 14.0 | **36.0** | 6.0 | 26.0 | 32.0 |
| **Shake Bottle** | 92.0 | **94.0** | 90.0 | 84.0 | 88.0 | 90.0 |
| **Stack Blocks Three** | 6.0 | **12.0** | 8.0 | 4.0 | 8.0 | 8.0 |
| **Stack Bowls Two** | 54.0 | **78.0** | 60.0 | 20.0 | 68.0 | 70.0 |
| **Stamp Seal** | 32.0 | 36.0 | 28.0 | 10.0 | **38.0** | **38.0** |
| **Avg** | 38.5 | 38.0 | 40.5 | 22.1 | 38.8 | **44.7** |

## D.2. LIBERO-Skill

**Task Setting**

*Table 9.* The task setting of LIBERO-Skill.

| Task | Skill | Task Instruction | Task Setting |
|------|-------|------------------|--------------|
| 0101 | pick place
pick place | Put both the alphabet soup and the tomato sauce in the basket. |  |
| 301 | open (prismatic)
pick place | Open the top drawer of the cabinet and put the bowl on the plate. |  |
| 401 | close (prismatic)
pick place | Close the top drawer of the cabinet and put the black bowl on the plate. |  |
| 014 | pick place
close (prismatic) | Put the black bowl in the bottom drawer of the cabinet and close the bottom drawer of the cabinet. |  |
| 46 | close (prismatic)
close (revolute) | Close the top drawer of the cabinet and close the microwave. |  |
| 32 | open (prismatic)
stack | Stack the middle black bowl on the back black bowl and open the top drawer of the cabinet. |  |
| 501 | open (revolute)
pick place | Put the black bowl on the plate and open the microwave. |  |

*Table 9.* The task setting of LIBERO-Skill (Continued).

| Task | Skill | Task Instruction | Task Setting |
|------|-------|------------------|--------------|
| 601 | close (revolute) pick place | Put the black bowl on the plate and close the microwave. |  |
| 74 | turn close (prismatic) | Close the drawer of the cabinet and turn off the stove. |  |

*Table 10.* Key parameters for Pre-Training on LIBERO-90.

| Parameter | Value |
|-----------|-------|
| *Training Config* | |
| Batch size | 32 |
| Total training steps | 20,000 |
| *Optimization* | |
| Peak learning rate | $2.5 \times 10^{-5}$ |
| Optimizer | AdamW |
| $\beta_1, \beta_2$ | 0.9, 0.95 |
| EMA decay | 0.99 |
| *Model Architecture* | |
| Action horizon | 10 |
| Number of experts | 4 |
| Top-k selection | 1 |
| Dimension of skill embeddings | 64 |

### D.3. In-Domain Tasks

**LIBERO.** LIBERO is a simulation benchmark with 130 different tasks. The standard task setting for in-domain evaluation contains four task suites: LIBERO-Spatial, LIBERO-Object, LIBERO-Goal, LIBERO-Long. Each suite consists of 10 different tasks, designed to evaluate capabilities on tackling variance on spatial layout, object identity, goal specification, and temporal composition.

**RoboCasa.** RoboCasa is a large-scale simulation framework designed to cultivate general-purpose robotic capabilities for daily living tasks. Featuring high-fidelity kitchen environments, the framework comprises over 2,500 3D assets and 100 distinct manipulation tasks. This evaluation benchmark utilizes a Panda robot integrated with an Omron gripper to rigorously assess household manipulation proficiency. The testing scope includes operating kitchen appliances, executing pick-and-place maneuvers, and interacting with articulated objects such as doors and drawers.

*Table 11.* Key parameters for training on LIBERO.

| Parameter | Value |
|---|---|
| *Training Config* | |
| Batch size | 128 |
| Total training steps | 30,000 |
| *Optimization* | |
| Peak learning rate | $2.5 \times 10^{-5}$ |
| Optimizer | AdamW |
| $\beta_1, \beta_2$ | 0.9, 0.95 |
| EMA decay | 0.99 |
| *Model Architecture* | |
| Action horizon | 10 |
| Number of experts | 4 |
| Top-k selection | 1 |
| Dimension of skill embeddings | 64 |

*Table 12.* Key parameters for training on RoboCasa.

| Parameter | Value |
|---|---|
| *Training Config* | |
| Batch size | 32 |
| Total training steps | 180,000 |
| *Optimization* | |
| Peak learning rate | $2.5 \times 10^{-5}$ |
| Optimizer | AdamW |
| $\beta_1, \beta_2$ | 0.9, 0.95 |
| EMA decay | 0.99 |
| *Model Architecture* | |
| Action horizon | 10 |
| Number of experts | 4 |
| Top-k selection | 1 |
| Dimension of skill embeddings | 64 |

*Table 13.* Evaluation results of SkillNet and baseline models on RoboCasa.

| Task | GR00T-N1.6 | $\pi_{0.5}$ | Vanilla MoE | SkillNet |
|---|---|---|---|---|
| **CloseDoubleDoor** | 88.5 | **90.0** | 86.0 | 86.0 |
| **CloseDrawer** | **100.0** | 98.0 | 96.0 | **100.0** |
| **CloseSingleDoor** | 96.0 | **100.0** | 98.0 | **100.0** |
| **CoffeePressButton** | **98.5** | 98.0 | 90.0 | 98.0 |
| **CoffeeServeMug** | 63.5 | 68.0 | **72.0** | 64.0 |
| **CoffeeSetupMug** | 31.0 | 26.0 | **42.0** | **42.0** |
| **OpenDoubleDoor** | 39.0 | **100.0** | 96.0 | 96.0 |
| **OpenDrawer** | 81.1 | 78.0 | **84.0** | 80.0 |
| **OpenSingleDoor** | 81.5 | 82.0 | 90.0 | **92.0** |
| **PnPCabToCounter** | 41.0 | 64.0 | 52.0 | **68.0** |
| **PnPCounterToCab** | 47.5 | **66.0** | 58.0 | 60.0 |
| **PnPCounterToMicrowave** | 19.0 | **34.0** | 20.0 | 26.0 |
| **PnPCounterToSink** | 46.0 | 58.0 | **74.0** | 62.0 |
| **PnPCounterToStove** | 63.2 | **72.0** | 70.0 | 70.0 |
| **PnPMicrowaveToCounter** | 24.5 | **52.0** | 46.0 | 40.0 |
| **PnPSinkToCounter** | 50.0 | 74.0 | 68.0 | **76.0** |
| **PnPStoveToCounter** | 54.5 | 72.0 | 74.0 | **78.0** |
| **TurnOffMicrowave** | 96.0 | 92.0 | **100.0** | 98.0 |
| **TurnOffSinkFaucet** | 93.5 | 94.0 | 98.0 | **100.0** |
| **TurnOffStove** | **31.0** | 18.0 | 22.0 | 28.0 |
| **TurnOnMicrowave** | 91.5 | 96.0 | 96.0 | **98.0** |
| **TurnOnSinkFaucet** | 89.0 | 90.0 | **94.0** | 92.0 |
| **TurnOnStove** | **76.5** | 70.0 | 60.0 | 60.0 |
| **TurnSinkSpout** | 87.0 | 72.0 | 78.0 | **90.0** |
| AVG | 66.2 | 73.5 | 73.5 | **75.2** |

# E. Details on Real-World Tasks

## E.1. Setup

To validate the performance of SkillNet, we conduct real-world experiments on a dual-arm humanoid robot, AgiBot G1, equipped with parallel grippers. Three camera views are used, mounted on the left wrist, right wrist, and the head, respectively. During both data collection and inference, the relative poses between the robot base and the tabletop are kept fixed to eliminate environmental variation.

For data collection, two VR controllers are employed to teleoperate the left and right arms independently. At each timestep, synchronized three-view RGB observations are recorded together with the corresponding dual-arm states and actions in joint space. All data are collected at a frequency of 30 Hz. Before use, we further filter the collected data by removing trajectories with excessive jitter and discarding static frames from the records, which helps reduce execution-time failures caused by getting stuck. Each task is collected with a target of 50 demonstrations, and the resulting number of trajectories for each training task is reported in the corresponding tables.

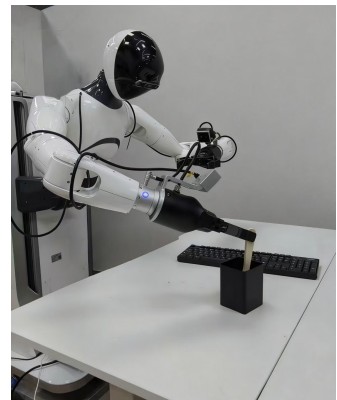

*Figure 9.* The real-world setup.

### E.2. Few-shot Adaptation

**Pre-trained Skills**

*Table 14.* The task setting of pre-trained skills on real-world environments.

| Skill | Traj.num | Task Instruction | Task Setting |
|---|---|---|---|
| cut | 49 | Pick up the knife and cut the playdough. | |
| draw | 50 | Pick up the pen and draw a wavy line on the whiteboard. | |
| erase | 149 | Pick up the blackboard eraser and erase the handwriting on the blackboard. | |
| flatten | 48 | Pick up the towel and flatten it. | |

*Table 14.* The task setting of pre-trained skills on real-world environments (Continued).

| Skill | Traj.num | Task Instruction | Task Setting |
|---|---|---|---|
| fold | 50 | Pick up the towel and straighten it up, pick up the towel and fold it twice. |  |
| hang | 48 | Pick up the clothes and hang them on the drying rack. |  |
| open (prismatic) | 147 | Open the upper drawer, put the item closest to the apple into the drawer, and close the upper drawer. |  |
| pick place | 30 | Pick up the lemon and place it on the plate on the right. |  |
| pour | 89 | Pick up the first cup from left to right and place it in the appropriate position and pour the coffee into the cup. |  |
| push | 49 | Push the cake next to the mineral water. |  |

*Table 14.* The task setting of pre-trained skills on real-world environments (Continued).

| Skill | Traj.num | Task Instruction | Task Setting |
|-------|----------|------------------|--------------|
| scoop | 49 | Pick up a spoon and scoop rice from the box into the bowl. |  |
| strike | 50 | Pick up a stick and strike the drum. |  |

**Transfer Skills**

*Table 15.* The task setting of transfer skills on real-world environments.

| Task id | Skill | Task Instruction | Task Setting |
|---------|-------|------------------|--------------|
| 1 | drag | Grab the traction rope of the small cart and drag it near the basket. |  |
| 2 | stamp | Pick up the seal and stamp it on paper. |  |
| 3 | wipe | Grab the towel and wipe off water stains on the surface of the object. |  |

*Table 15.* The task setting of transfer skills on real-world environments (Continued).

| Task id | Skill | Task Instruction | Task Setting |
|---------|-------|------------------|--------------|
| 4 | hang | Pick up the clothes hanger and hook it onto the clothesline. |  |
| 5 | shake | Shake the mineral water bottle. |  |
| 6 | sweep | Pick up the broom to sweep garbage into the container. |  |
| 7 | brush | Grab the brush and use it to brush the surface of the keyboard. |  |
| 8 | insert | Pick up the straw and insert it into the bottle. |  |
| 9 | click | Click any key on the keyboard twice. |  |

*Table 16.* Key parameters for the Real-World Transfer experiments. Both pre-training and fine-tuning on transfer tasks use a multi-task training strategy.

| Parameter | Value | |
|---|---|---|
| | **Pretrain** | **Transfer** |
| *Training Config* | | |
| Batch size | 32 | 32 |
| Total training steps | **30,000** | **20,000** |
| *Optimization* | | |
| Peak learning rate | $2.5 \times 10^{-5}$ | $2.5 \times 10^{-5}$ |
| Optimizer | AdamW | AdamW |
| $\beta_1, \beta_2$ | 0.9, 0.95 | 0.9, 0.95 |
| EMA decay | 0.99 | 0.99 |
| *Model Architecture* | | |
| Action horizon | 50 | 50 |
| Number of experts | 4 | 4 |
| Top-k selection | 1 | 1 |
| Dimension of skill embeddings | 64 | 64 |

### E.3. Zero-shot Generalization on Task Compositions

Similar to the simulator settings, we design 21 pre-trained tasks with diverse skill compositions, along with 6 unseen task compositions, to evaluate zero-shot generalization to novel compositions. The training tasks are constructed based on a set of 21 distinct skills described in Appendix E.2, which encompasses both the pre-trained and transfer skills in the transfer experiments.

*Table 17.* Settings of real-world compositional tasks with zero-shot evaluation.

| Task | Skill | Task Instruction | Task Setting |
|---|---|---|---|
| Drag+Click | drag click | Grab the traction rope of the small cart and drag it near the basket, then click any key on the keyboard once. |  |
| Brush+Drag | brush drag | Grab the brush and use it to brush the surface of the keyboard, then grab the traction rope of the small cart and drag it near the basket. |  |
| Sweep+Shake | sweep shake | Pick up the broom to sweep garbage into the container, then shake the mineral water bottle. |  |

*Table 17.* Settings of real-world compositional tasks with zero-shot evaluation (Continued).

| Task | Skill | Task Instruction | Task Setting |
|---|---|---|---|
| Wipe+Click | wipe click | Grab the towel and wipe off water stains on the desktop, then click any key on the keyboard once. |  |
| Open+Close | open (prismatic) pick place close (prismatic) | Open the upper drawer, then put the bread into the drawer, then close the drawer. |  |
| Drag+Shake | drag shake | Grab the traction rope of the small cart and drag it near the basket, then shake the mineral water bottle. |  |

*Table 18.* Key parameters for pre-training before zero-shot evaluation on unseen task compositions.

| Parameter | Value |
|---|---|
| *Training Config* | |
| Batch size | 32 |
| Total training steps | 30,000 |
| | |
| *Optimization* | |
| Peak learning rate | $2.5 \times 10^{-5}$ |
| Optimizer | AdamW |
| $\beta_1, \beta_2$ | 0.9, 0.95 |
| EMA decay | 0.99 |
| | |
| *Model Architecture* | |
| Action horizon | 50 |
| Number of experts | 4 |
| Top-k selection | 1 |
| Dimension of skill embeddings | 64 |

## E.4. Long-horizon Tasks

*Table 19.* Real-world long-horizon tasks under in-domain settings.

| Task | Skill | Task Instruction | Task Setting |
|------|-------|------------------|--------------|
| Bru+Clk | brush click | Grab the brush and use it to brush the surface of the keyboard, then click any key on the keyboard once. |  |
| Cut+Stp | cut stamp | Pick up the knife and cut the playdough, then pick up the seal and stamp it on paper. |  |
| Op+PP+Cls | open (prismatic) pick place close (prismatic) | Open the upper drawer, then put the blackboard eraser into the drawer, then close the drawer. |  |
| PP+Drg | pick place drag | Put the chewing gum into the small cart, then grab the traction rope of the small cart and drag it near the basket. |  |
| PP+Shk | pick place shake | Pick up the cube and put it into the cup, then shake the cup. |  |
| PP+Str | pick place stir | Pick up the cube and put it into the cup, then pick up chopsticks and stir the contents of the cup. |  |

*Table 19.* Real-world long-horizon tasks under in-domain settings (Continued).

| Task | Skill | Task Instruction | Task Setting |
|------|-------|------------------|--------------|
| Swp+Wip | sweep wipe | Pick up the broom to sweep garbage into the container, then pick up a towel and wipe the desktop. |  |

*Table 20.* Key parameters for training on in-domain long-horizon tasks.

| Parameter | Value |
|-----------|-------|
| *Training Config* | |
| Batch size | 32 |
| Total training steps | 30,000 |
| | |
| *Optimization* | |
| Peak learning rate | $2.5 \times 10^{-5}$ |
| Optimizer | AdamW |
| $\beta_1, \beta_2$ | 0.9, 0.95 |
| EMA decay | 0.99 |
| | |
| *Model Architecture* | |
| Action horizon | 50 |
| Number of experts | 4 |
| Top-k selection | 1 |
| Dimension of skill embeddings | 64 |

## F. Analysis on Efficiency

To evaluate efficiency, we measured the parameter, memory usage, and inference FLOPs for $\pi_0$, $\pi_{0.5}$, $\pi_{0.5(MoE)}$, and SkillNet. The detailed comparisons, with $\pi_{0.5}$ serving as the baseline, are presented in Table 21. We observe that while SkillNet increases the parameter count and memory usage by 27.2% and 24.7% respectively compared to the $\pi_{0.5}$ baseline, the increase in inference FLOPs is significantly lower at 11.8%. This demonstrates that SkillNet is well-suited for real-time robotic manipulation tasks. Taking a standard NVIDIA RTX 4090 GPU ($\approx$83 TFLOPS) as an example, the baseline $\pi_{0.5}$ (1.86T FLOPs) theoretically supports a control frequency of approximately 45 Hz. In comparison, SkillNet (2.08T FLOPs) maintains a robust frequency of roughly 40 Hz. Given that typical robotic control loops operate at 10–20 Hz, SkillNet effectively balances expanded capacity with the efficiency required for high-frequency real-time control.

*Table 21.* Efficiency comparison (overhead is relative to $\pi_{0.5}$)

| Model | Parameters | | Memory Usage | | Inference FLOPs | |
|-------|------------|---------|--------------|---------|-----------------|---------|
| | Value | Overhead | Value | Overhead | Value | Overhead |
| $\pi_0$ | 3.24B | – | 6.60GB | – | 1.74T | – |
| $\pi_{0.5}$ | 3.35B | – | 6.80GB | – | 1.86T | – |
| $\pi_{0.5(MoE)}$ | 4.26B | +27.2% | 8.48GB | +24.7% | 2.08T | +11.8% |
| SkillNet | 4.26B | +27.2% | 8.48GB | +24.7% | 2.08T | +11.8% |

