# OpenReview forum: "SkillNet: Hierarchical Skill Modeling for Compositional Generalization in Vision-Language Action Models"
_ICML.cc/2026/Conference — ICML 2026 regular_

### Official Review · Reviewer_t97j · 2026-03-12

**Soundness:** 3
**Presentation:** 3
**Significance:** 3
**Originality:** 3
**Overall Recommendation:** 4
**Confidence:** 4

**Summary:**

This paper addresses two major shortcomings of existing skill-centric approaches—namely, their loosely organized structures lacking hierarchical organization and their inability to represent transferable skill attributes within a structured parameter space—by proposing the SkillNet model. SkillNet hierarchically models diverse skills and leverages a mixture-of-experts framework to represent transferable skill attributes. The paper reports experimental results under few-shot and zero-shot settings, demonstrating the effectiveness of the proposed method from multiple perspectives.

**Compliance With Llm Reviewing Policy:**

Affirmed.

**Final Justification:**

The rebuttal addressed my concerns, and I maintained my positive score.

**Key Questions For Authors:**

See strength and weakness

**Limitations:**

See strength and weakness

**Strengths And Weaknesses:**

Strength
1，The authors decompose skills into three hierarchical levels—mechanical attributes, semantic categories, and specific skill instances—and represent them through unified learnable embeddings. Compared with unstructured skill tokens or fully implicit skill discovery methods, this structured organization provides stronger interpretability and systematic representation.
2，The Skill-Contextualized MoE integrates skill sequences into routing decisions through cross-attention, while incorporating shared experts and a router balance loss to ensure stable training. This design enables skill similarity to be mapped to similarity in expert routing paths, thereby supporting compositional generalization.
3，The experimental design is relatively comprehensive, and is further supported by representation similarity analysis, resulting in overall well-substantiated empirical validation.

Weakness
1，	The motion code taxonomy and VerbNet category selection mentioned in the paper still rely on the authors’ specific manual choices and the distribution of the dataset. When new atomic skills emerge, it remains unclear whether the introduction of new data would cause instability in the model’s performance. In addition, it is unclear whether the model’s capability for compositional skill generalization might negatively affect its original capabilities. Furthermore, when encountering more complex or ambiguously defined instructions—such as “organize the desk,” which may involve multiple actions including moving, stacking, and opening—it is uncertain whether this fixed hierarchical representation can adequately adapt.

2，	SkillNet introduces additional components, and although the authors show that inference cost and memory requirements remain manageable, the construction of hierarchical skill modeling relies on large-scale offline data annotation (over 470K demonstration samples). Therefore, it is unclear whether the construction cost of the method outweighs the performance gains it provides. Moreover, the paper lacks deeper discussion on the generalizability of the method and the cost of extending it to new domains.

Q:

1.Although this paper many different experiments, but some composite tasks seem strange, like wipe the desk and click the keyboard. They seems there are no relation. Why do you design such composition?
2.How do you model the hierarchy structure of different skills?
3.You mention some skill-centric methods in the related works. Additional comparison experiments between your method and the skill-centric methods will improve the persuasiveness.
4. During the training process, how can we ensure that the routing mechanism of Mixture of Experts (MoE) has good generalization when facing unseen actions? How can we ensure that the optimal skill combination is selected when a task can be completed by a variety of different action combinations? Could this selection mechanism fail in long-term sequence tasks because there is currently no module related to memory mechanisms?

---

> ### Author Rebuttal · Authors · 2026-03-31
>
> Thanks for your constructive suggestions! We provide rebuttals on weakness and questions below.
>
> **W1&Q4: Robustness and generalization on unseen skills. How to tackle ambiguous instructions.**
>
> When unseen skills emerge, SkillNet identifies and extends it to the most relevant node on semantic and motion patterns with top-down search in HSM. In addition, a new skill node initialized randomly is added for each unseen skill in the last $c_R$ layer, to generate a node path and corresponding contextual embedding. This process enables the model’s generalization to unseen behaviors during training. This is validated on few-shot experiments for unseen skills (Sec.4.2). Regarding stability and catastrophic forgetting when introducing new data, we conducted additional evaluations on RoboTwin-2.0. After training on 15 distinct tasks and subsequent fine-tuning on a novel skill, SkillNet demonstrated a 12.3% lower performance degradation (forgetting rate) compared to $\pi_{0.5}$. This indicates that SkillNet achieves better stability and robustness through hierarchical skill modeling.
>
> As mentioned in Line 205-208 (right), ambiguous instructions are decomposed into subtasks and confirmed with visual inputs before execution using VLMs. Thus, a novel skill 'organize' is already decomposed into 'opening', 'stacking', or 'moving' before execution. Meanwhile, SCMoE is agnostic to the underlying backbone and can be readily extended to incorporate memory modules, making it a promising direction for future work.
>
> **W2: The construction cost. Deeper discussion on generalizability to new domains.**
>
> We first analyzed the cost on offline data annotation. Our records indicate that each sample requires approximately 1.5K tokens, costing around 0.0016 dollars using GPT-5.1-mini. In total, annotation on 470K demonstrations consumed roughly 710M tokens, costing less than $800. In comparison, based on the scaling law, the improvements of SkillNet on zero-shot (16%) and few-shot tasks (23.9%) are equivalent to 10-50 times of data scaling during pre-training. Thus, the cost to construct SkillNet is substantially lower than increasing the training data to get such improvements.
>
> For generalization to new domains, SkillNet provides a scalable foundation for continuous expansion. Two core components of HSM (motion code and VerbNet classes) are universal for motion representations. When encountering novel embodiments or unfamiliar physical environments (e.g., extraterrestrial environments), the motion codes can be seamlessly supplemented with additional encoding positions rather than from-scratch annotations of existing categories.
>
> **Q1: Explanation to task compositions.**
>
> We design this task ('Wipe' + 'Click') for scenarios where a user cleans the desk before working or typing. Moreover, these compositions under zero-shot settings are intentionally designed to be challenging and evaluate the model’s generalization capabilities. This ensures that the model can be evaluated on real OOD settings and helps assess whether the model can compose previously learned atomic skills in novel ways, rather than overfitting to skill sequences during training.
>
> **Q2: Hierarchy of different skills.**
>
> We model the hierarchy of different skills with three-layer abstractions from motion categories $c_M$, semantic categories $c_V$, to specific skill realizations $c_R$. The motion category $c_M$ is informative on mechanical properties, using motion code to capture attributes on contact, deformation, and etc. Built on top of this, the semantic category $c_V$ with VerbNet classes refines motion categories $c_M$ at a higher level by incorporating semantic distinctions. In particular, it differentiates skills based on their targets and thematic roles. Finally, the lowest level $c_R$ corresponds to specific skill realizations expressed as verb phrases. Each $c_R$ is grounded in a concrete action description and is organized under its corresponding semantic category $c_V$. We provide details of hierarchical skill modeling in Line 193-202.
>
> **Q3: Additional comparison with skill-centric methods.**
>
> We first provide additional comparisons with API-based methods (e.g., CaP). We reproduce this pipeline using GPT-5.1-mini for task planning. On LIBERO-Skill, this pipeline achieves 38% success rate on a relatively simple task “0101”, while SkillNet achieves 74% on the same task. These methods are still limited by the capability of individual APIs and error accumulation across multiple calls.
>
> We further compare with AtomicVLA (arXiv, Mar 2026), a concurrent work more closely aligned with our settings. On LIBERO-Skill, SkillNet largely outperforms AtomicVLA by 31.4% in a zero-shot manner. Meanwhile, on in-domain settings, SkillNet also outperforms AtomicVLA (99.1% versus 97.8% on LIBERO). This stems from hierarchical skill modeling and the soft transition mechanism from SCMoE which models transitions between skills to reduce overfitting on specific skill compositions.

---

> > ### Author Rebuttal · Reviewer_t97j · 2026-04-03
> >
> > Most of concerns have been adequately addressed

---

> > > ### Author Response · Authors · 2026-04-06
> > >
> > > Thank you for your recognition of the novelty and experimental design! We will incorporate the responses from this rebuttal into the revised manuscript.

---

### Official Review · Reviewer_6Nhr · 2026-03-12

**Soundness:** 3
**Presentation:** 4
**Significance:** 4
**Originality:** 3
**Overall Recommendation:** 5
**Confidence:** 3

**Summary:**

This paper proposes SkillNet, a skill-centric Vision-Language Action (VLA) framework designed to tackle the challenges of compositional generalization and few-shot transfer to unseen behaviors in robotic manipulation. The core of SkillNet consists of two main components: 1) Hierarchical Skill Modeling (HSM), which explicitly structures skills across multiple levels of abstraction, from physical mechanics (via Motion Codes) to semantics (via VerbNet) and specific skill realizations; 2) Skill-Contextualized Mixture-of-Experts (SCMoE), an architecture that leverages the hierarchical skill attributes as soft constraints to route the MoE network. This design forces the network to share parametric pathways for skills with similar physical and semantic attributes. The authors evaluate SkillNet extensively in both simulation environments (LIBERO, RoboTwin, RoboCasa) and real-world dual-arm robot setups, demonstrating significant improvements over state-of-the-art flow-matching VLAs (e.g., π0.5, GR00T) in zero-shot task compositions and few-shot unseen behavior transfers.

**Compliance With Llm Reviewing Policy:**

Affirmed.

**Key Questions For Authors:**

Robustness to Label Noise: Your automatic data curation pipeline (Appendix B) relies on a VLM to extract verb phrases and motion codes. How sensitive is the SCMoE routing to tagging errors? Could you provide a brief ablation study where you inject a certain percentage (e.g., 10%, 20%) of incorrect motion/semantic tags during inference to see how gracefully the model's performance degrades?
Interpretability of Soft Transitions: In Section 3.3, you mention that cross-attention allows the model to "softly attend to different skill stages." Could you provide a visualization of the cross-attention weights over the skill sequence E_((S1:T))  across the timesteps of a real-world execution (e.g., a "Pick -> Place -> Close" task)? This would strongly validate your claim that the model knows when to transition.
Deep Dive into Failure Cases: In Table 2, the "Wipe + Click" zero-shot task yields a very low success rate (15.0%). Why does SkillNet struggle specifically with this combination? Is it due to the drastic difference in force dynamics (continuous friction vs. discrete pressing) causing representation conflicts in the shared experts?
Fairness in Efficiency/Capacity Comparison: According to Table 21, SkillNet has 27.2% more parameters than the dense π0.5 baseline. While the Vanilla MoE comparison is helpful, what happens if you scale up the dense π0.5(increasing its depth/width) to exactly match the 4.26B parameter count of SkillNet? Does the performance gap persist?

**Limitations:**

The authors briefly mention the "optimal depth of such hierarchies" as an open question in the Conclusion, and provide a generic, boilerplate Impact Statement. However, they do not adequately discuss the concrete limitations of their method. I suggest adding a dedicated "Limitations" paragraph to discuss: 1) The reliance on the accuracy of external VLMs for skill tagging; 2) The manual engineering effort required to define and maintain the Motion Code taxonomy and VerbNet subsets, which might hinder effortless scaling to completely open-world, in-the-wild robotic datasets.

**Strengths And Weaknesses:**

Soundness:
Strengths: The paper is technically highly sound. The empirical evaluation is extensive and rigorous, covering zero-shot, few-shot, and in-domain settings across multiple simulators and a real-world humanoid robot. The inclusion of a "Vanilla MoE" baseline is highly commendable, as it successfully isolates the performance gains of the HSM-guided routing from the mere increase in model capacity. Furthermore, the theoretical justification in Appendix A (bounding the policy Total Variation distance by attribute divergence) and the Representational Similarity Analysis (RSA) in Appendix C provide excellent, rigorous evidence that the routing strategy genuinely aligns with human-defined semantic topologies.
Weaknesses: The pipeline relies heavily on an external, lightweight VLM (GPT-5-mini) for the automatic data curation of Motion Codes and VerbNet tags. This introduces a potential vulnerability: the paper lacks an analysis of how robust the SCMoE routing is to label noise or VLM hallucination during inference. Additionally, while the paper claims that cross-attention over the skill sequence E_((S1:T))enables "soft transitions" between skills without hard boundaries (Eq. 4), there is no qualitative evidence (e.g., attention map visualizations) demonstrating how the model internally tracks the temporal progress of multi-stage tasks.Significance: Does the paper address an important or relevant problem? Does it advance understanding, capabilities, or practice in machine learning? Could it influence future research or applications (e.g., other researchers or practitioners are likely to use the ideas or build on them)? Is the scope of impact broad or specialized, and is that appropriate for the contribution? Even if the improvements are modest or domain-specific, could they unlock new directions or provide practical utility?

Presentation:
Strengths: The paper is exceptionally well-written and easy to follow. The motivation is compelling, and the progression from the problem (lack of structured skill sharing) to the solution (HSM + SCMoE) is logical. Figures 1, 2, and 3 are of high quality and effectively illustrate the complex concepts of hierarchical modeling and MoE routing.
Weaknesses: The authors report failure cases honestly (e.g., the 15% success rate on "Wipe + Click" in Table 2), but they fall short of providing a deep dive or qualitative analysis in the main text to explain why these specific skill compositions fail catastrophically.

Significance:
Strengths: The paper addresses a critical bottleneck in the current VLA scaling paradigm: the inability to generalize to unseen task compositions without exponential data scaling. By injecting a strong, physics- and linguistics-inspired inductive bias into the MoE routing, this work provides a highly impactful new direction for robot learning. The findings will likely inspire future works on structured parameter sharing in embodied AI.

Originality:
Strengths: The originality lies in the highly creative synthesis of existing tools. While Motion Codes (Paulius et al.), VerbNet, and MoE are existing concepts, combining them to force a neural network to learn a compositional parametric space that mirrors a physical/semantic hierarchy is a novel and brilliant perspective in the context of Flow-Matching VLA models.

---

> ### Author Rebuttal · Authors · 2026-03-31
>
> Thanks for your constructive suggestions! We provide rebuttals on weakness and questions below.
>
> **Q1&W1: How sensitive is the SCMoE routing to tagging errors?**
>
> During evaluation on LIBERO-Skill, we randomly replaced skill tokens with incorrect motion/semantic tags at different probabilities to validate the robustness to tagging errors. The results show that even with 20% of skill tokens corrupted, SkillNet’s performance only degrades by 4.0% on average, while still maintaining a 12% improvement over the baseline ($\pi_{0.5}$). This indicates that SCMoE routing is relatively robust to moderate tagging errors.
> | Error injection percentage (\%) | 0101 | 301 | 401 | 014 | 46 | 32 | 501 | 601 | 74 | avg |
> |-----|-----|-----|-----|-----|-----|-----|-----|-----|------|------|
> | 5 | 70.0 | 60.0 | 92.0 | 64.0 | 22.0 | 26.0 | 20.0 | 42.0 | 18.0 | 46.0 (-1.8) |
> | 10 | 72.0 | 52.0 | 90.0 | 60.0 | 28.0 | 28.0 | 18.0 | 42.0 | 18.0 | 45.3 (-2.5) |
> | 20 | 66.0 | 48.0 | 90.0 | 62.0 | 24.0 | 26.0 | 16.0 | 48.0 | 14.0 | 43.8 (-4.0) |
>
> Furthermore, we conducted an error analysis on the outputs generated by the VLM. Specifically, we selected non–pick-and-place demonstrations that involve ambiguous instructions to focus on challenging edge cases. Across 200 samples, we found that, within the motion codes derived from annotations, there were no instances of completely incorrect encodings. Only 7% of the cases exhibited minor, acceptable deviations (e.g., slight discrepancies in object rigidity/plasticity annotations). Comparatively, our experiments with incorrect tag injection indicate that such deviations have a limited and acceptable impact on overall performance.
>
> **Q2&W2: Visualization of transitions on a real-world execution.**
>
> We provide a qualitative visualization ([Link1](https://anonymous.4open.science/r/rebuttal-icml-574F/router-plot.pdf)) of the expert routing dynamics during a multi-stage task: Pick $\rightarrow$ Brush $\rightarrow$ Click. The routing patterns exhibit clear, phase-aligned shifts from Pick Phase, Brush Phase, to Click Phase. Different expert combinations exhibit regular alternations during the execution phase aligned with skills. Meanwhile, we observe progressive transitions in routing patterns across different skill stages. These routing signals offer a direct and interpretable view of the transition logic with attention weights as inputs.
>
> **Q3: Deep dive into failure cases.**
>
> Thank you for your careful observation and constructive feedback. This is exactly caused by the drastic difference in force dynamics between skills ('wipe' with continous friction using tools and 'click' with temporal contact with target). Meanwhile, this task also represents for a class of tasks demanding longer transition distance between different skill executions. Previous methods didn't include specialized designs to handle such boundary cases. To address this, we experimented with an intuitive improvement: when there is a large dynamic discrepancy during skill transitions, we reset the robot’s end-effector to its initial pose before executing the next stage. With this adjustment, SkillNet’s performance on the "Wipe + Click" task improves to 35.0%.
>
> **Q4: Fairness in efficiency/capacity comparison.**
>
> We fully expanded the action expert of $\pi_{0.5}$ using the same approach as SkillNet, keeping the total number of parameters unchanged. We then evaluated its zero-shot performance on LIBERO-Skill. As shown in the table below, the expanded $\pi_{0.5}$ shows only a marginal improvement over the original version and still falls significantly short of SkillNet. This confirms that the improvements are not primarily driven by parameter scaling, but by effective sharing and switching among experts.
>
> | Model | Full parameters (B) | 0101 | 301 | 401 | 014 | 46 | 32 | 501 | 601 | 74 | avg |
> |-----------|---------------|------|-----|-----|-----|----|----|-----|-----|----|------|
> | $\pi_{0.5}$ (extend)  | 4.26B        | 54.0 | 48.0 | 86.0 | 42.0 | 28.0 | 16.0 | 14.0 | 30.0 | 0.0 | 35.3 |
> | SkillNet     | 4.26B            | 74.0 | 56.0 | 92.0 | 60.0 | 28.0 | 24.0 | 20.0 | 56.0 | 20.0 | 47.8 (+12.5) |
>
> **Limitations:** Thanks for your constructive suggestions. As the volume of embodied data grows, hierarchical data organization becomes more and more important, as embodied tasks naturally exhibit structural dependencies compared with pure language understanding tasks. The effort on design and construction of HSM makes an early attempt on skill data hierarchy and provide a basis with universal motion and semantic categories. Expansion on HSM can be completed with additional encoding positions on motion codes without from-scratch annotation on in-the-wild dataset. Furthermore, with advancements in techniques like VLM caching, the cost of scaling for such hierarchical structures will be further mitigated. We will incorporate these discussions into our revised manuscript.

---

> > ### Author Rebuttal · Reviewer_6Nhr · 2026-04-05
> >
> > Thank you for the author's reply.

---

> > > ### Author Response · Authors · 2026-04-06
> > >
> > > Do you have additional questions on our work that were not adequately resolved in our rebuttal? We are very happy to further clarify them.

---

### Official Review · Reviewer_DFit · 2026-03-13

**Soundness:** 3
**Presentation:** 3
**Significance:** 3
**Originality:** 3
**Overall Recommendation:** 4
**Confidence:** 3

**Summary:**

This paper proposes a method for addressing long-horizon manipulation using skills. Existing skill-based approaches have faced challenges in how to hierarchically organize skills and how to use them in a parametric way within the model. To address this, the paper proposes SkillNet, which extracts mechanical properties using motion codes and semantic properties using VerbNet, and uses them to hierarchically organize skills. This hierarchical model is referred to as HSM. The paper then incorporates HSM into a skill-conditioned MoE, where the hierarchical information is distributed across different experts through skill-conditioned routers at each layer so that the skills are well represented in the MoE structure. Through this design, the method enables compositional behavior through skill reuse, and shows good performance on zero-shot tasks with unseen skill orders, few-shot versions of such settings, and in-domain generalization.

**Compliance With Llm Reviewing Policy:**

Affirmed.

**Final Justification:**

Many of the concerns have been adequately addressed, and after considering the rebuttal, I will maintain my score.

**Key Questions For Authors:**

- When curating motion codes with a VLM, if the VLM labels a motion code using the image at timestep $t$ and the instruction from an episode, is that motion code paired with the chunk of actions from $t:t+c$, that is, the action chunk of size $c$?
- In Figure 2, does the layer embedding refer to the activation from the previous layer?
- In Equation 5, the text suggests that the previous layer activation is refined through cross-attention with the skill embedding, but in the figure, cross-attention seems to be applied only for generating the router weights. Which interpretation is correct? Also, what does $\phi$ refer to?
- For $c_R$ , the paper states that “bottom level skill realizations $c_R$ are grounded in demonstrations from DROID and AgiBot-World, covering over 200 distinct verbal phrases.” It is still unclear what this grounding process exactly means. More concretely, how is $c_R$ learned and constructed?

**Limitations:**

yes

**Strengths And Weaknesses:**

### Strength

- The paper compares against the necessary VLA baselines and evaluates on relevant benchmarks, which supports the effectiveness of SkillNet.
- The benchmark evaluation is systematic across zero-shot, few-shot, and in-domain generalization settings, and the description of these settings is very clear.
- The paper also introduces LIBERO-Skill, which can evaluate unseen subtask orders in the original LIBERO benchmark.
- The paper also presents real-world experiments with well-designed evaluation settings and a substantial number of tasks.
- The hierarchical modeling through HSM is carefully constructed by combining motion codes and VerbNet-based semantics, and substantial effort appears to have gone into this design.
- The hierarchical skill-conditioned MoE shows good performance, and the overall idea is quite intuitive.
- The paper includes various ablations, including SCMoE vs. vanilla MoE vs. a pure model, as well as failure studies on LIBERO-Skill with π0, π0.5, and SkillNet, which support the necessity of SkillNet and SCMoE.

### Weakness

- The main claim is that the model builds reusable skills and improves performance through their efficient reuse. However, although the paper makes this claim, it would be helpful to provide more evidence from an efficiency perspective to support it more clearly.
- It would also be helpful to provide qualitative visualizations showing how a specific skill is selected for a given input, how the expert gates change over the course of an episode, and which experts and skills are actually reused across different tasks.
- It would also be helpful to analyze how previous skill-based models structured their skills, including through hierarchy or other forms of organization, and to clarify why the proposed method is more effective than those alternatives. In addition, since the process of constructing HSM appears somewhat complex, the paper should better explain whether there are relevant prior works or prior knowledge motivating this design choice, and why this design is well justified.

---

> ### Author Rebuttal · Authors · 2026-03-31
>
> Thanks for your constructive suggestions! We provide rebuttals on weakness and questions below.
>
> **W1: An efficiency perspective on skill modeling.**
>
> (1) Task modeling efficiency: Our method uses a compact skill vocabulary to support a combinatorially large task space. On LIBERO-Skill, 8 distinct skills are targeted to cover 90 tasks in the training set and generalize to 9 unseen compositions, whereas using only language modeling would require 99 different task templates. The number of tasks represented by skill compositions also grows exponentially with skill count and the sequence length, and we use reusable skills to degrade this complexity.
>
> (2) Data efficiency: SkillNet improves sample efficiency by leveraging shared structures across skills. The Hierarchical Skill Model captures skill similarity, and SC-MoE enables routing pattern sharing among related skills. As shown in Figure ([Link1](https://anonymous.4open.science/r/rebuttal-icml-574F/few-shot-transfer.png)), with only 15 demos, SkillNet surpasses $\pi_{0.5}$ trained on 25, and with 25 demons, it exceeds $\pi_{0.5}$ trained on 50. The heatmap ([Link2](https://anonymous.4open.science/r/rebuttal-icml-574F/skill-router-similarity.png)) further shows strong alignment between skill similarity and routing patterns, indicating effective expert reuse across related skills. Quantitative analysis is provided in Line 726–751.
>
> **W2: Qualitative visualizations for how gates or skills changed and reused.**
>
> In Figure ([Link3](https://anonymous.4open.science/r/rebuttal-icml-574F/router-plot.pdf)), we provide a qualitative visualization of how expert gating changes over an episode. For a representative task (pick–brush–click), we observe clear stage-wise shifts in dominant experts, with smooth transitions enabled by SC-MoE. Moreover, we analyze the relationship between skill transitions and routing dynamics in Line 370-384 (right).
>
> In Figure ([Link4](https://anonymous.4open.science/r/rebuttal-icml-574F/task-heat-map.pdf)), we illustrate how both skill primitives and motion categories are reused across 15 real-world tasks. We observe that most skills are reused multiple times across different tasks. Through compositional reuse, these skills can be combined to support a combinatorially large space of tasks, enabling the emergence of novel task variations.
>
> **W3: How previous methods structured skills. Better explain on prior knowledge.**
>
> Using skills as the fundamental execution units for manipulation tasks is a relatively new topic and has attracted increasing attention. Previous works focus on the utility of skills on long-horizon task decomposition. To the best of our knowledge, most of them organize skills in a flat manner, where each skill is associated with a semantic label and implemented independently. Our key insight is how to organize these independent skills and enable compositional generalization. Inspired by how ImageNet organize visual entities with WordNet, we first adopt VerbNet to capture semantic relationships among skills. Furthermore, analogous to objects being composed of attributes and parts, we argue that skills also require decomposition into more fundamental units or mechanical properties. This motivates moving beyond flat or purely semantic structures toward a compositional hierarchy. In our approach, motion code provides a practical mechanism for such decomposition.
>
> **Q1: Intervals on motion code curation.**
>
> During data curation, we segment demonstrations into variable-length intervals using keyframes detected via velocity minima. Each motion code is associated with the segment between two consecutive keyframes, resulting in temporally adaptive, semantically meaningful units rather than fixed-size chunks.
>
> **Q2: Reference for layer embedding.**
>
> The layer embedding corresponds to the activation from the previous layer.
>
> **Q3: Position of cross-attention. Meaning of $\phi$.**
>
> Cross-attention is applied only for generating the router weights. We find it more effective to use skill embeddings as routing conditions, which improves compositional generalization in zero-shot and few-shot settings. $\phi$ denotes an MLP projection applied before cross-attention, which maps both the skill embeddings and the layer activations into a shared feature space with aligned dimensionality.
>
> **Q4: How is $c_R$ learned and constructed.**
>
> $c_R$ is extracted from demonstrations via a three-step automatic pipeline. For each demonstration, we first extract keyframes at speed minima to reduce sequence length while preserving mechanical transition points. Second, with these keyframes and the task instructions, we employ an VLM (gpt-5.1-mini) to decompose the process into a sequence of fine-grained subtasks. Finally, for each subtask, we extract verb phrases that describe the core actions, while filtering out irrelevant information. The resulting verb-centric representations are then used to form $c_R$. Details are provided in Line 663–703.

---

> > ### Author Rebuttal · Reviewer_DFit · 2026-04-04
> >
> > Most of my concerns have been addressed. However, regarding the qualitative visualization of the router, it is still unclear whether each expert’s activation corresponds to any distinct semantic meaning. I understand that such interpretation is not straightforward, but from the authors’ perspective, do particular experts appear to respond to specific skills? Do the authors believe there may be any interpretable relationship here?

---

> > > ### Author Response · Authors · 2026-04-06
> > >
> > > Thank you for the insightful question. Rather than manually assigning an expert for each skill, we model the relationship between skills and expert combinations in a compositional manner. Each skill is represented by a combination of expert activations, which improves modeling efficiency and enables structural sharing across skills. Therefore, individual experts are not expected to encode distinct skills. Instead, we expect to observe consistent expert compositions aligned with skills.
> > >
> > > To support this, we performed Representational Similarity Analysis (RSA) between expert activation patterns and skill representations (Line 708-751). The results suggest that expert activation patterns capture meaningful structure aligned with skill representations. Similar or identical skills exhibit high similarity in their expert composition, while different skills show clear distinctions. A more straightforward visualization is shown in the heatmap ([Link2](https://anonymous.4open.science/r/rebuttal-icml-574F/skill-router-similarity.png)), where similarities of skills on semantic and mechanical properties are reflected in the similarity of their expert combinations.
> > >
> > > In addition to this compositional structure at the skill level, we further examine whether individual experts exhibit preferences over lower-level motion patterns. We observe that structured relationships emerge at the level of more fundamental mechanical properties (motion codes). For example, on the first expert layer, for skills involving short-term forceful contact (e.g. click, strike, stamp), we found that expert_3 is more frequently activated. For skills considering long-term contact with tool-using (e.g., brush, wipe, sweep, scoop), expert_0 is more frequently activated. For skills with repeated actions (e.g., brush, sweep, strike), expert_2 are more frequently activated. To validate this observation, statistical analysis across 20 episodes for relationships between representative skills and expert activations are provided in the table below.
> > >
> > > | Skill | expert_0 | expert_1 | expert_2 | expert_3 |
> > > |-----|-----|-----|-----|-----|
> > > | click | 0.02 $\pm$ 0.03 | **0.54** $\pm$ 0.08 | 0.07 $\pm$ 0.06 | **0.37** $\pm$ 0.08 |
> > > | strike | 0.00 $\pm$ 0.00 | 0.06 $\pm$ 0.04 | **0.31** $\pm$ 0.06 | **0.63** $\pm$ 0.05 |
> > > | stamp | 0.00 $\pm$ 0.00 | **0.32** $\pm$ 0.07 | 0.17 $\pm$ 0.07 | **0.51** $\pm$ 0.09 |
> > > | brush | **0.33** $\pm$ 0.08 | 0.26 $\pm$ 0.10 | **0.40** $\pm$ 0.08 | 0.01 $\pm$ 0.02 |
> > > | scoop | **0.65** $\pm$ 0.06 | 0.14 $\pm$ 0.07 | 0.12 $\pm$ 0.04 | 0.09 $\pm$ 0.05 |
> > > | sweep | **0.39** $\pm$ 0.09 | 0.20 $\pm$ 0.11 | **0.28** $\pm$ 0.05 | 0.13 $\pm$ 0.03 |
> > >
> > > This is a very meaningful open question regarding interpretability when introducing inductive biases into MoE routing mechanism, which worth further exploration in the future. In our approach, such correspondences are not explicitly enforced, but rather emerge from the inductive bias during learning, and are therefore not expected to be strict. Overall, these results suggest that the relationship between experts and skills is structured and consistent, emerging at the level of expert compositions and underlying mechanical properties.

---

### Decision · Program_Chairs · 2026-04-30

**Decision:**

Accept (regular)

**Comment:**

This paper was reviewed by three experts in the field and received ratings of two Weak Accept ratings and an Accept rating. Based on the reviewers’ feedback, the decision is to recommend the paper for acceptance to ICML 2026. Reviewers viewed the paper positively overall, highlighting the broad and systematic evaluation across zero-shot, few-shot, in-domain, and real-world settings, the inclusion of a strong vanilla-MoE baseline, and the intuitive combination of hierarchical skill modeling with skill-contextualized MoE routing. In particular, reviewers found the paper’s central idea interesting: organizing skills through motion codes, VerbNet semantics, and skill instances, then using these structured skill representations to guide expert routing and support compositional generalization in VLA models. The rebuttal also helped strengthen confidence by addressing questions on routing interpretability, robustness to noisy tags, and fairness of the capacity comparison.

Overall, the paper is a good contribution on an important problem. In the final version, we ask the authors to incorporate the relevant points from the discussion and rebuttal into the paper, as these clarifications would further strengthen the presentation and practical impact of the work. We congratulate the authors on the acceptance of their paper.